# Preparation and Use of Decellularized Extracellular Matrix for Tissue Engineering

**DOI:** 10.3390/jfb13040240

**Published:** 2022-11-14

**Authors:** Adam D. McInnes, Michael A. J. Moser, Xiongbiao Chen

**Affiliations:** 1Division of Biomedical Engineering, College of Engineering, University of Saskatchewan, Saskatoon, SK S7N 5A9, Canada; 2Department of Surgery, Health Sciences Building, University of Saskatchewan, Saskatoon, SK S7N 0W8, Canada; 3Department of Mechanical Engineering, College of Engineering, University of Saskatchewan, Saskatoon, SK S7N 5A9, Canada

**Keywords:** 3D printing, biomaterials, decellularization, ECM, extracellular matrix, recellularization, regenerative medicine, scaffolds, tissue engineering

## Abstract

The multidisciplinary fields of tissue engineering and regenerative medicine have the potential to revolutionize the practise of medicine through the abilities to repair, regenerate, or replace tissues and organs with functional engineered constructs. To this end, tissue engineering combines scaffolding materials with cells and biologically active molecules into constructs with the appropriate structures and properties for tissue/organ regeneration, where scaffolding materials and biomolecules are the keys to mimic the native extracellular matrix (ECM). For this, one emerging way is to decellularize the native ECM into the materials suitable for, directly or in combination with other materials, creating functional constructs. Over the past decade, decellularized ECM (or dECM) has greatly facilitated the advance of tissue engineering and regenerative medicine, while being challenged in many ways. This article reviews the recent development of dECM for tissue engineering and regenerative medicine, with a focus on the preparation of dECM along with its influence on cell culture, the modification of dECM for use as a scaffolding material, and the novel techniques and emerging trends in processing dECM into functional constructs. We highlight the success of dECM and constructs in the *in vitro*, *in vivo*, and clinical applications and further identify the key issues and challenges involved, along with a discussion of future research directions.

## 1. Introduction

Regenerative medicine and tissue engineering are related and overlapping multidisciplinary fields that seek to restore health for patients through promoting a remodelling and repair process. Though the terms are often used synonymously, and often combined under the umbrella of tissue engineering and regenerative medicine, there is a distinction between these two disciplines. Regenerative medicine is a field within medical science that uses cells, scaffolds, growth factors and other signalling molecules, and/or gene manipulation to restore or establish normal function through endogenous healing to repair, regenerate, or replace cells, tissues, or organs through *in vivo* and *ex vivo* techniques. Tissue engineering lies within the overlap between regenerative medicine and biomedical engineering, and uses techniques that combine scaffolds, cells, and biological materials to to manufacture and grow new tissues *in vitro* and *in vivo* that repair or replace diseased, damaged, or missing tissue or organs. Together, these fields seek to provide cures to complex and often chronic diseases rather than treatments that manage disease [1,2].

Scaffolds play a key role in tissue engineering, where the biomaterial used to construct the scaffolds is central. These biomaterials, either natural or synthetic, must be biocompatible and biodegradable and possess characteristics appropriate or favourable for the regeneration of the tissue type by serving as a temporary support in place of the native extracellular matrix (ECM), though they are later replaced by new-regenerated tissue. Some form of the ECM can be found in all tissues of living multicellular organisms. Plants make use of cellulose, the most abundant biopolymer on the planet, to construct their cell walls [3]; fungi use chitin for their cell walls (as do arthropods in their exoskeleton); and all animals use a combination of proteoglycans, polysaccharides, glycoproteins, and proteins to form their ECM. Throughout human history, ECM has found extensive use in the form of leather, sinew, catgut, etc. in manufacturing tools, instruments, and clothing. More recently, scientists have begun to recognize the importance of the ECM in tissue formation and development, and thus a potential for use in tissue engineering and regenerative medicine. With the compounds in ECM, decellularized ECM (dECM) has the ability to stimulate a remodelling and repair response *in vivo* by inducing an M2 macrophage response rather than an M1 macrophage response with inflammation and fibrosis [4,5,6,7,8]. Researchers have also found that dECM can promote a more stable cell phenotype [9,10,11,12,13,14]. Over the past decade, considerable progress has been made in the development of dECM for tissue engineering and regeneration. Here, we review this progress with an emphasis on the structure and properties of the ECM, methods to prepare dECM, methods to modify and process dECM into functional constructs, the *in vitro*, *in vivo*, and clinical applications of dECM constructs, followed by a discussion of the recommended research directions. 

## 2. Structure and Properties of ECM

The ECM is a complex system of molecules that play a vital role in the body. The paradigm has begun to shift towards seeing the functional unit as being the cell and the ECM surrounding it, due to the importance of the ECM for cell function [15]. This is because the ECM provides mechanical and biochemical signals to the cells it surrounds, and though produced by the cells themselves, the complex heterogeneous matrix allows cells to polarize and to assume the appropriate phenotype in a concept known as “dynamic reciprocity” [15,16].

The ECM is, by mass, mostly a protein-based structure in tissues that acts as a skeleton to support and hold cells in place. This core matrisome contains over 300 types of proteins, primarily collagens, and includes other proteins and protein-based molecules. Broadly, these molecules are classified as structural proteins, such as collagen and elastin; adhesive proteins, such as fibronectin and laminin; and ground material, in the form of various glycoproteins and proteoglycans [17,18]. In the larger matrisome, which is both the ECM proteins and ECM-associated proteins and factors (grouped into six broad categories: collagens, ECM-affiliated proteins, proteoglycans, ECM glycoproteins, ECM regulators, and secreted factors), there are approximately 1200 different proteins [18,19,20]. By weight, collagen is the most abundant protein in the body, and has a triple helix structure that forms a collagen molecule; such molecules join to form an interlocking microfibril, with the microfibrils joining to form fibrils (Figure 1).

The larger matrisome includes matricellular proteins. These are proteins that are found in the pericellular matrix region of the ECM (not to be confused with the pericellular space that is present in cartilage, which is encapsulated in the territorial matrix). Like other ECM proteins, the proteins of the pericellular matrix are secreted by cells into their environment, but the matricellular proteins serve little to no structural role, though they might bind to or affect structural proteins. Instead, these proteins help to regulate the behaviour of cells by influencing cell–matrix interaction, bind to cell receptors in an autocrine or paracrine manner, activate or sequester various cytokines, proteases, and other bioeffector molecules [16,21,22]. 

The ECM is more complicated than simply the proteins and protein-based molecules that comprise the ECM and the matricellular environment. Within the ECM, enzymes, miRNA, growth factors, and other cell signalling molecules are stored in the ECM in nanovesicles [23,24,25]. In addition, the ECM contains cryptic epitopes and crypteins that can have similar or vastly different effects than their parent protein [26,27,28,29,30]. These compounds primarily have effects as tissues are degraded, such as in tissue remodelling and inflammation in response to injuries.

### 2.1. Xenogeneic Compatibility of ECM

The molecules that comprise the ECM play fundamental roles in physiology, so their amino acid sequence and structure are generally highly conserved throughout evolution and the divergence of species. Material detected in dinosaur fossils, notably *Tyrannosaurus rex* (MOR 1125, 66 Ma), *Brachylophosaurus canadensis* (MOR 2598, 80 Ma), and *Lufengosaurus* (CXPM Z4644, 195 Ma), is very suggestive of collagen I based on multiple methods of analysis. Analysis of the amino acid sequences suggests that there is a strong evolutionary relationship between dinosaurs and both avian and crocodilian species [31,32,33,34]. More specifically, amino acid sequencing of fragments of collagen α1(I) and α2(I) from *T. rex* has demonstrated a strong evolutionary relationship to modern *Gallus gallus* and *Struthio camelus* (the common chicken and the common ostrich, respectively) with a 0.90 confidence in Bayesian analysis in the generated phylogenetic tree [35]. Furthermore, from analysis across 14 extant species representing ∼450 Ma of Vertebrata evolution, the sequence of the coding regions (exons) of the COL1A1 gene, an ortholog coding for two of the three subunits in collagen I, has been demonstrated to be highly conserved evolutionarily among vertebrates [36]. *In vivo* use of xenogeneic dECM from mammalian tissue in humans is effective in part due to the highly conserved proteins and ligands in the ECM across species, especially those species that are more closely related through evolution [37,38].

### 2.2. Immune Response to dECM

The immune response to allogeneic and xenogeneic dECM is very different than the immune response to whole allogeneic and xenogeneic tissue grafts, and generally leads to greater success of the decellularized graft. Similar to the immune response seen following the implantation of any foreign object into the body, the dECM is rapidly invaded by immune cells of several different types. These immune cells begin to degrade the dECM, breaking it down to peptides, creating and releasing crypteins, exposing cryptic epitopes, and releasing the contents of dECM nanovesicles, which triggers a remodelling rather than an inflammatory response. 

There are several different immune cells, called leukocytes, that respond to and invade the dECM. The leukocytes that are primarily responsible for responding to foreign bodies and antigens are histocytes and lymphocytes. Histocytes are derived from circulating monocytes, and differentiate into dendritic cells, macrophages, and Langerhans cells within tissues. In the immune response, histocytes have a role in activating lymphocytes (T and B cells) and regulating the inflammatory response.

Of the histocytes involved, macrophages play a key role in determining the immune response to dECM, by adopting an inflammatory or anti-inflammatory response. There are two main phenotypes of macrophages seen *in vitro*: M1 and M2. A simplified view is that M1 macrophages are responsible for mediating a pro-inflammatory response and fibrosis, while M2 macrophages induce an anti-inflammatory response that promotes repair, remodelling, and cell proliferation, but this paradigm has shifted. There are four different subtypes of M2 macrophages, M2a, M2b, M2c, and M2d [4,39,40]. An emerging distinction is being recognized between how macrophages respond to dECM compared to the traditional macrophage profiles, with the macrophage phenotype having a unique gene expression profile, and markers of this dECM-stimulated macrophage being termed “M_ECM_” [24,41]. There are also hybrid macrophages that can exhibit markers of both M1 and M2 macrophages. However, this *in vitro*–defined macrophage paradigm is not necessarily reflective of what is seen in the in vivo immune response. *In vivo*, there are macrophages of different polarizations and activation markers that coexist in tissues and often display different markers than what are seen *in vitro* [42,43]. Nevertheless, the M1/M2 paradigm is still useful in understanding the *in vitro* and *in vivo* responses to implanted biomaterials, and the phenotypes of macrophages seen *in vivo* are often referred to as M1-like and M2-like cells for this reason.

The tissue source of the dECM can influence the polarization of macrophages and the phenotype ratios of recruited macrophages. *In vitro*, the phenotype of the macrophage response varies with dECM based on the source of the tissue: gastrointestinal, brain, and urinary dECM stimulates a strong M2 response; liver and skeletal muscle dECM does not polarize the macrophage ratio to either M1 or M2 phenotypes; and in the presence of dermal dECM, a predominantly M1 population develops. Considering the M_ECM_ paradigm, the response to dECM and the M_ECM_ marker profile that develops depends on the tissue source of the dECM, with suggestions that a more accurate phenotype definition would be “M_‘source-tissue’-ECM_” [24,41]. Careful selection of the tissue source for the dECM is important, as the source of the dECM has the potential to stimulate specific macrophage phenotypes to develop *in vitro* that are analogous to macrophages *in natura* [7,8,41,44].

The other important leukocyte to consider in the immune response to dECM is the thymocyte (T cell). Of the multiple types of T cells, the two CD4^+^ effector lineages of T-helper cells, Th1 and Th2, are important to the type and degree of the immune response that is seen to dECM. Each cell type has a specific role in the immune response: Th1 cells are responsible for cellular immunity, fighting intracellular problems such as cancer and viruses, and are associated with graft rejection; and Th2 cells are responsible for humoral immunity by mediating the production of antibodies to extracellular pathogens and are associated with immune tolerance. Properly decellularized ECM elicits very little Th1 response, and the immune response that is seen to decellularized xenogeneic grafts is identified with a Th2 cytokine profile. This is demonstrated by the initial response to implanted decellularized xenogeneic grafts with an early, short-duration inflammatory response followed by organized, site-appropriate tissue remodelling and repair. Though T cells play a role in the immune response to implanted ECM, they do not appear to play any role in the constructive tissue remodelling process that is seen with dECM [37,45,46].

Following implantation, there is a strong immune response to the dECM. Within a week, dECM scaffolds are typically infiltrated with an abundant mononuclear population with a smaller neutrophil response. The monocyte population typically has a high M2:M1 ratio of macrophages, which is consistent with a remodelling and repair response with limited inflammation. The macrophages have an important role in regulating the dECM degradation, assembly, and remodelling response [47,48]. Neutrophils, though present early in the immune response, appear to play only a minor role in the remodelling process of dECM. The body quickly starts to form new ECM to replace the ECM in the scaffold. During the remodelling process, by two weeks there is a strong chemotaxis response that stimulates host cells to migrate into the dECM scaffold and start growing new tissue, and there is a strong angiogenesis stimulus that causes vasculature to develop throughout the scaffold [48,49,50,51]. Adding stem cells to dECM prior to implantation have a synergistic effect with the ECM of activating the immune response towards an M2 phenotype response [52].

Due to supply, availability, and restrictions, much of the research done on ECM has been conducted using xenogeneic ECM. Of concern with xenogeneic tissues is the presence of the α-Gal epitope, Galα1-3Galβ1-4GlcNAc-R. The α-Gal epitope is found on cells and in the ECM of all non-primate mammals, New World monkeys, and some microbes in the normal human gut flora, but humans and Old World monkeys lack these epitopes. As a result, the human immune system responds with a very strong inflammatory response to xenogeneic tissues with this epitope, which precludes the possibility of successful xenogeneic tissue transplants. Though cells are removed from tissues during decellularization, and most of the α-Gal epitopes are removed during decellularization, the process does not completely remove these epitopes from the dECM. However, xenogeneic dECM has been in clinical use since the late 1990s—and for thousands of years if you consider catgut and sinew sutures [53,54]—and of the several million recipients, some of whom have received multiple dECM implants, there have been no reported incidents of rejection or even sensitization to the α-Gal epitope present in these dECM implants [37] Furthermore, use of dECM from α-Gal-knockout porcine has not been shown to improve clinical outcomes or have any effect on the immune response when compared to dECM from wildtype porcine [37]. There are methods of removing the α-Gal epitopes remaining in dECM, such as α-galactosidase or PNGase F treatment, but they are cost prohibitive, and are generally not used due to lack of improved clinical outcomes compared to untreated dECM [37,55]. Together, these findings suggest that the presence of α-Gal in the quantities found in ECM is not a serious problem for the use of xenogeneic dECM in tissue engineering.

The nature of the immune response to implanted materials is important to consider for clinical applications. In inflammatory responses, macrophages can form multinucleated giant cells, and in the presence of materials that cannot be phagocytosed or otherwise eliminated, multinucleated giant cells will wall-off the foreign material and induce fibroblasts to engage in fibrosis and scar formation. This response is known as a foreign body response and is often seen with synthetic implants or slowly degrading biomaterials, which can be a desirable response in securing implants but can be undesirable if scar formation occurs in unintended and harmful ways. Acellular ECM induces an anti-inflammatory response *in vivo* in macrophages that triggers remodelling and repair, reducing scar formation and promoting the growth of new, site-appropriate, and functional tissues [6,7,56,57]. This type of response is favourable for implants where the formation of scar tissue is counterproductive to healing and where growth of new tissue is desirable. 

### 2.3. Cryptome and Nanovesicles

The influence of ECM on cells is further complemented in the repair and remodelling of tissues by factors that are released. As dECM is degraded in tissue remodelling and repair, functional molecules are created through the breakdown of proteins as well as through the release of enzymes, growth factors, and other cell signalling molecules stored in the ECM in nanovesicles [23,58]. During degradation of the dECM, these proteins break down to release peptides that promote cell migration, proliferation, and remodelling, and play a role is modulating biological processes such as angiogenesis and the inflammatory response [23,26,27,28,29,59,60].

Cryptic peptides and cryptic epitopes are important classes of peptides, known as matrikines, that comprise the cryptome and influence cell physiology. Cryptic peptides, also known as crypteins, are not complete proteins in and of themselves but rather are bioactive protein fragments that have different physiological effects than their parent protein. Crypteins differ from cryptic epitopes, also known as matricrypteins, in that cryptic epitopes are hidden or sequestered where they are not seen by cells, but are revealed as a result of an inflammatory response that induces changes in the protein conformation and plays a role in the immune and healing response [61,62]; in contrast, crypteins are revealed through proteolysis due to various physiological and pathological process, and they have targeted physiological effects. Crypteins are typically divided into three classes [26,27,28]: Class 1—peptides proteolytically cleaved *in vivo* that are novel and function very differently than their parent protein;Class 2—peptides proteolytically cleaved *in vivo* and have similar activity to their precursors; andClass 3—peptides produced *in vitro* through proteolytic digestion of proteins or recombinant technology, but may not be similar or identical to those found *in vivo*.

ECM-derived crypteins, also known as matricrypteins, are produced specifically though degradation and remodelling of tissues, such as wound healing, the pathological response to tumour growth, and the immune response [26,27,28,29,59]. Endogenous enzymes, such as matrix metalloproteinases, elastase, etc., cleave the ECM proteins into peptides, some of which have properties of cryptic peptides. These matricrypteins have various biological effects, including chemotaxis, adhesion, and anti-microbial effects as well as in both angiogenesis and anti-angiogenesis [37]. These hidden bioactive sequences have effects *in vivo*, and they may be useful *in vitro* for tissue-engineering research as well as having potential for novel pharmaceutical applications; see Table 1.

Within the ECM, enzymes, miRNA, growth factors, and other cell signalling molecules are stored in the ECM in nanovesicles that are known as matrix-bound nanovesicles (MBV). These matrix-bound nanovesicles have a distinct profile from the exosomes and microvesicles that are produced by a wide variety of cells and found in various extracellular fluids. These nanovesicles, ranging in size from 10 to 1000 nm, are found in the extracellular matrix of soft tissues and lack identifiable markers on their membrane, and though there are some common components, the content of these nanovesicles differs between the different tissue sources for the ECM. MBVs have also been shown to play an important role in macrophage activation pathways in response to dECM scaffolds as well as in the differentiation of stem cells. Moreover, isolated MBVs have been shown to have similar effects to dECM on cells and, in some cases, a more potent effect [23,24,25,63]. 

**Table 1 jfb-13-00240-t001:** Select crypteins of the ECM.

Parent ECM Protein	Protein Chain	Cryptein Name	MW (kDa)	Purpose in Parent Protein	Function	References
Collagen III	IIIα	AGVGGEKSGGF	~1	C terminus telopeptide	Chemotactic behaviour Increases the presence of Sox2+ and Sca1+, Lin− cells at wound site Influences osteogenesis and bone remodelling	[64,65]
Collagen IV	α_1_	Arresten	26	NC1 domain	Inhibits angiogenesis (inhibits endothelial cell proliferation, migration, and tube formation) Inhibits tumour growth and metastasis	[66]
	α_2_	Canstatin	24	NC1 domain	Inhibited endothelial cell proliferation and migration Endothelial cell apoptosis	[67]
	α_3_	Tumstatin	28	NC1 domain	Inhibits angiogenesis (amino acids 54–132) Promotes adhesion and inhibit proliferation of human melanoma cells (amino acids 185–203) Inhibit proliferation, promote apoptosis, and inhibit Akt activation (amino acids 185-191; CNYYSNS linear peptide) Reduces neovascularization (YSNSG cyclopeptide)	[68]
Collagen XV	α_1_	Restin	22	NC1 domain	Anti-angiogenic Tumour-growth inhibition	[28]
Collagen XVIII		Endostatin	20	NC1 domain	Inhibit angiogenesis Inhibits in vivo growth of primary and metastatic tumours	[28,69,70]
Perlecan		Endorepellin	81	C terminus	Blocked adhesion of endothelial cell to fibronectin and type I collagen Binds and counter-acts endostatin	[71,72]
Fibronectin	III_1C_	Anastellin	10.18	C terminus two-thirds of the first type III homology repeat	Suppress tumour growth and metastasis Inhibit angiogenesis Affects cell cycle progression	[73]
Laminin-332	γ2	EGF-like repeat	30	DIII	Stimulate cell migration without proliferation	[74,75]
Laminin-111	β1	β1–LN–LE1-4 fragment	60	N terminus	Regulates cell behaviour (e.g., epithelial-to-mesenchymal transition) Downregulates MMP2 expression	[76]
Elastin		xGxPGxGxG consensus sequence	~0.75		Stimulate cell migratory, proliferative, and morphogenic behaviours Stimulates angiogenesis Pro-tumour properties	[77,78,79,80,81]

### 2.4. Effects on Cell Behaviour

Seeding cells into dECM is a common technique used for *in vitro* and *in vivo* experiments. The most common sources of these cells are primary cells, cell lines, and stem cells. Stem cells have the unique ability to differentiate into different tissue types, but they generally require specialized treatments to induce this differentiation into a particular cell type. The specific make-up of the ECM is unique to each tissue type, and the ECM has been shown to have the potential to induce and enhance the differentiation of stem cells towards the phenotype associated with the tissue source of the ECM [14,56,82,83,84,85,86,87,88,89,90,91,92,93,94,95,96,97,98,99,100,101,102,103,104,105].

The ECM is able to promote a more stable cell phenotype. Cells have been shown to be affected by the physical and biochemical cues in their milieu. These cues trigger different intracellular responses that direct physiological responses within the cell. For some cell types, the ECM is crucial to proper cell function. As an example, hepatocytes and sinusoidal endothelial cells are very sensitive and difficult to culture *in vitro*, rapidly losing their phenotype and function. Given that the liver is well known for having a high regenerative capacity, including the ability to grow in a matter of weeks to replace the tissue transplanted in partial liver transplants and to regenerate in cases of other types of liver damage due to the rapid *in natura* proliferation of hepatocytes and sinusoidal endothelial cells, it seems surprising that these cell types are difficult to culture *in vitro*. Culturing hepatocytes and sinusoidal endothelial cells on dECM has been shown to prolong phenotype retention and increase cell function compared to culturing in collagen I, and this effect is greatly enhanced by culturing these cells on dECM derived from livers [9,10,11,12,13,106]. Kidney dECM has been shown to be more effective at promoting the recruitment and migration of host cells, accelerate the formation of vascular networks and maintain vascular integrity, and contribute to the self-arrangement and maturation of cells to form glomerular-like structures *in vivo* [98]. Fibroblasts have been shown to be more responsive to their microenvironment when grown in skin dECM bioink compared to type I collagen [107]. Cartilage dECM has been shown to be effective at preventing chondrocyte hypertrophy and calcification of the cartilage in the repair of cartilage defects [108]. The ability to improve cell viability and function and promote cell proliferation has also been shown in other cell types when cultured on dECM and, especially, tissue-specific dECM [14,56,82,93,98,109,110,111,112,113,114,115].

The ECM can also have other effects on cell culture. The make-up of ECM changes with age, and these differences can affect the behaviour of cells in culture, including ECM from younger animals having the capacity to rejuvenate aged mesenchymal stem cells so that the aged cells regain many of the key properties seen in younger cells, and to alter cell behaviour in injury repair [116,117,118,119,120,121,122]. The species from which the dECM is derived will also influence the behaviour of cells due to the species-dependent biologic cues intrinsic to the ECM [123]. The stiffness of the substrate can also provide cues for stem cell differentiation and influence fibrosis in injury repair [122,124]. Different tissue sources of the dECM can also affect the behaviour of cell lines in tissue culture, with widely different behaviours of cells being reported [125]. Culturing cells in dECM derived from tumours can cause cells to behave in similar patterns to cancer cells, including increased proliferation, migration, invasion, and stimulation of angiogenesis [126,127,128,129]. Using dECM from diseased tissue, such as cirrhotic livers, can shift cell behaviour from what is seen in healthy tissue [130]. Regional differences in ECM composition from a given tissue, such as regions of the meniscus, can influence cell proliferation and the mechanical properties of the ECM produced by cells growing in the scaffolds [131]. Culturing cells *in vitro* in growth media previously cultured in dECM or supplemented with extracts from dECM has also been shown to promote cell proliferation, viability, and differentiation and stimulate anti-inflammatory macrophage phenotypes [23,86,91,92,132,133,134]. Cells have been shown to develop markers and morphologies that indicate more mature phenotypes when cultured in dECM [93,135,136,137,138]. Together, these results demonstrate the importance of selecting the appropriate tissue source for dECM to achieve the best outcomes [125,139].

ECM can regulate cell growth and differentiation, and one proposed mechanism for how cells to sense and respond to the stiffness ECM through mechanotransduction is the tensegrity architecture model hypothesis. This model hypothesizes that cells stabilize their shape and sense mechanical signals through the integrins binding to the ECM, which in turn stresses the cytoskeleton and creates a buckling, soft-strut tensegrity [140,141,142,143]. Though there is still debate about the validity of the tensegrity architecture model versus the continuum model (i.e., a viscous cytoplasm surrounding the viscous cytoplasm and the elastic nucleus), the model potentially provides at least part of a multi-factorial explanation of cell mechanotransduction response to ECM, influencing the remodelling and repair response.

## 3. Methods of Preparing dECM

Vertebrate ECM can be derived from various tissues by means of different methods. One common step, which is frequently used among these methods, is to freeze the tissues to cause cell rupture and separation of the tissue layers. Decellularization involves mechanical and chemical methods to remove cells from tissues and retain the intact ECM structures. Early methods of decellularization involve the use of chemical baths. These methods work well for simple, thin tissues, such as bladders and digestive organs, but not for intact solid organs. Effective decellularization protocols for solid organs involve cutting the organ into thin slices or small pieces and decellularizing them in a chemical bath, but this disrupts the architecture of the organs and prevents the use of the resulting dECM as an intact structure. To have an intact solid organ involves perfusing the vasculature tree (and other anatomical structures such as airways [123]) with decellularizing fluids to disrupt and remove the cells evenly across the tissues. Perfusion of the vascular tree also works for other simple and complex tissues where it is desirable to have an intact structure [144,145].

Detergents are the most common treatment used to decellularize tissues. This is because detergents are effective at disrupting the phospholipid membrane of cells, which allows the cell and its contents to be removed from the tissue. However, detergents can negatively affect cells in the recellularization of dECM. Sodium dodecyl sulfate (SDS) is arguably the most common detergent used for decellularization. In an optimization study with SDS, Friedrich EE et al. (2017) reported that the residual amount of SDS left in the dECM was sufficient to cause an increased inflammatory response and fibrosis, both *in vitro* and *in vivo*, and also reported using CaCl_2_ to precipitate SDS from the dECM (81.4% vs. 98.4%) [146]. Ghorbani F et al. (2021) demonstrated that residual SDS, even the low residual SDS concentrations often found in ECM decellularized using SDS, detected using a methylene blue assay to detect the anionic surfactant, results in decreased cell viability based on the resistance of the cell phenotype to SDS [147]. He M et al. (2017) conducted an optimization study using SDS at different concentrations and durations of exposure and found that effective decellularization could be completed even at low concentrations and shorter durations of exposure, and that shorter exposure times were associated with better retention of both structural and functional ECM biomolecules [148]. In a study by Kawasaki T et al. (2015) comparing SDS with another detergent, SDS was found to damage the ECM microstructure, destroy the ECM laminar array, remove most of the sulfated glycosaminoglycans (sGAG), and affect the growth factors and cytokines [149]. Weng et al. (2021) reported that cell-cultured bone ECM decellularized using SDS was cytotoxic to cells during recellularization [103]. Furthermore, Uhl FE et al. (2020) demonstrated that the decellularization of lungs using SDS resulted in high losses of GAGs, and the remaining GAGs were dysfunctional and unable to bind key matrix-associated growth factors [150]. These findings suggest that SDS is not an ideal detergent for decellularization.

Triton X-100 is comparable to SDS in prevalence of use for decellularization. While Triton X-100 is not as effective at removing antigenic cell components as SDS that can lead to an inflammatory response *in vivo*, Triton X-100 is generally less damaging to the extracellular matrix structure and has a better retention rate of bioactive molecules than SDS [151]. As a result, Triton X-100 is often used in combination with and to complement other detergents and methods to effectively remove cells and cell debris from the ECM. There are many other detergents, chemicals, enzymes, and methods that are used to decellularize tissues. More information on these is available in Table 2.

One of the important considerations for selecting decellularization techniques is quality control, ensuring that sufficient decellularization has been achieved, so as to not interfere with experimental results. Proposed metrics have been established for the minimum standards for achieving adequate decellularization, and these have largely been adopted. Analysis of decellularization efficiency can then be assessed using a combination of three proposed standard criteria:nuclear material not visible in tissue sections stained with either H&E or 4′,6-diamidino-2-phenylindole (DAPI);dsDNA content < 50 ng/mg of ECM (dry weight); andDNA fragment size < 200 base pairs [233,234]

These minimal criteria have been established in order to reduce the intensity of any immune response to the foreign tissues [5]. Mora-Navarro et al. (2022) established a protocol using absorbance spectroscopy that allowed them to monitor the decellularization progress in real-time, by measuring the absorbance of the DNA content in the effluent at 260 nm [235]. Though these are established criteria, they might not be suitable for all tissues, and there may be a need to consider some of the cellular components in a full standard for adequate decellularization, such as the levels of residual mitochondria, phospholipids, MHC-1 proteins, α-Gal epitopes, etc. At this time, these metrics serve as the gold standard for ensuring adequate decellularization. 

### 3.1. Decellularization in Chemical Baths

The simplest methods of decellularization involve the use of chemical baths. For the simplest tissues, this can often be achieved by a chemical bath. Tissues such as urinary bladders and gallbladders, small intestines, and joint capsules can be processed by removing all the tissue layers and leaving the lamina propria, submucosa, and synovium, respectively, and washing them with a weak solution of peracetic acid and ethanol [23,82,92,128,158,195,196,197,200,223,224,225,227,230]. 

Solid organs and cartilage can also be decellularized by washing in chemical baths, but they generally need to be cut into thin slices or small pieces to ensure adequate, even, and rapid exposure to the decellularizing chemicals. This technique also requires more complex treatments, such as detergents and nucleases, to achieve adequate decellularization [83,110,113,114,116,125,159,161,162,171,182,211,236,237]. Skin, blood vessels, some mucosal tissues, cardiac valves, trachea, nerves, and pericardium often require a combination of treatments used for simple tissues and solid organs, requiring the mechanical removal of unwanted connective tissues layers, washes with detergents, and often treatment with a nuclease to more effectively degrade the genetic material [23,93,109,128,151,158,160,166,167,170,179,180,183,184,187,198,200,230]. Dense tissues such as tendons can be decellularized primarily through a series of freeze–thaw cycles followed by treatment with nucleases, though detergents can be added if the tissue is properly prepared (e.g., minced) [86,132,134,155,215].

Some tissues require additional treatment as a part of decellularization. Adipose and occasionally bone must have lipids removed as part of the process. Common solvents that are used for this process include isopropanol, acetone, hexane, methanol, ethanol, and diethyl ether [56,82,87,112,199,236]. Bone has to go through the added step of demineralization to extract the ECM, and this is generally accomplished by treating the bone with HCl [20,181,195,236]. 

### 3.2. Decellularization by Perfusion

Using dECM sheets works well for simple organs, such as hollow organs or skin. Unfortunately, they do not work so well with larger structures such as solid organs. Solid organs are too large and complicated in structure to be assembled from stacks of dECM sheets. Solid organs cannot be decellularized the same way that hollow tissues can—they cannot be soaked in solutions of chemicals and enzymes, and the solution cannot be expected to permeate through the tissues before the tissue spoils. Slicing the organ into thin strips does allow for simple decellularization; however, those slices cannot be used as a tissue scaffold on which to grow a full organ. Perfusion decellularization of solid organs and even simple tissues has proven to be effective.

The earliest successful work in perfusion decellularization was the work of Harald Ott. In 2008, he was training with Doris Taylor at the University of Minnesota, to find a way to decellularize solid organs. Realizing that the vascular tree was the simplest method of perfusing a solid organ because essentially every cell has a blood vessel next to it, Ott tried various enzymes and chemical solutions, but these either destroyed the ECM along with the cells or caused the tissues to swell and damage the ECM. Continuing to try other chemical solutions, Ott tried a common detergent, SDS, and perfused a rat heart through the native vasculature. This detergent was successful, washing away the cells and leaving the ECM intact down to the cellular scale, which included keeping the vascular tree intact [238].

Following this early success, Ott HC et al. developed a process where they perfused the hearts with heparinized phosphate buffered saline (PBS) which contained adenosine to flush the blood from the tissue and to dilate the blood vessels [239], respectively, followed by a 1% solution of SDS for 12 h to rupture the cell membranes and remove the cellular contents from the organ. This was followed by flushing via perfusion with deionized water, then a second perfusion with a 1% solution of Triton X-100 that helped to remove any remaining SDS and cell debris and renature the remaining proteins [240]. To maintain the decellularized scaffold and prevent bacterial growth, they perfused the “ghost organ” with PBS, to which they added an antimicrobial cocktail. Through this process, Ott HC et al. created a complex, anatomically accurate cardiac ECM scaffold with an intact and patent vascular tree and functional valves [241].

This was just the beginning of this research group’s work in this area. In 2010, Ott HC et al. demonstrated the effective decellularization of lungs using similar techniques [242]. Continuing their efforts, in 2013, Song J et al. focused their efforts on another organ, the kidney, adapting their previous work on hearts and lungs [243]. Pushing the boundaries of decellularizing and recellularizing scaffolds even further, in 2015, Jank BJ et al. began working on composite tissue regeneration of soft tissues. This team focused on a rat forelimb and a non-human primate limb. Following previous organ protocols, the limb was amputated and decellularized with detergents and PBS, though fasciotomies were performed prior to perfusion to allow the muscle areas to clear cell debris [145]. In 2018, Gerli MFM et al. succeeded in decellularizing a human arm using the perfusion protocols established in their previous work [244].

Following the success of Ott et al. with the perfusion decellularization of organs, this technique has become a popular method for decellularizing complex tissues. This method is even used when the dECM will be intentionally disrupted in further processing after decellularization. There are a variety of approaches to perfusing different tissues, even with the same tissues. The heart can be perfused through the coronary arteries by a retrograde perfusion of the aorta [137,138,241], or the superior or inferior vena cava [214]. The lungs are generally decellularized through the vasculature, by canulating and perfusing the pulmonary arteries or the main artery segment that branches into a lobe [117,242], though the lungs provide an additional approach of adding decellularizing solutions in addition to perfusing the vasculature: the airways can also be inflated with decellularizing solutions [123]. For many solid organs and tissues, canulating and perfusing the primary artery is one of the best approaches. The kidney is perfused through the renal artery [148,164,243], the liver through the portal vein and/or the hepatic artery [169,245], the pancreas through the splenic vein [99], the uterus through the uterine artery [153], and the corpus cavernosum through the cavernosal artery [157]. Other organs that are typical decellularized through chemical baths can be decellularized if there is a need to keep the structure intact, such as perfusing the superior mesenteric artery to decellularize the jejunum [144], the common carotid artery for the cervical esophagus [246], and the ureter for bladders [185]. Generally, the decellularizing fluid is filtered and recirculated through the organ, and the solution is changed as needed, but such a recirculation is not always possible [228].

### 3.3. Cell-Cultured ECM

In addition to obtaining the ECM by decellularizing tissues, the ECM can be grown using cell cultures. This is how commercial products such as Matrigel, Geltrex, and Humacyte’s Human Acellular Vessels are produced. The essential steps are providing a substrate for cells to grow, followed by providing the necessary nutrients and physiological conditions, and the the cells will produce the ECM that can be decellularized and used. This can achieve simple sheets of dECM that can be used for other experiments or procedures [101,103,126,209,247,248,249,250,251,252,253,254,255,256,257], the modification of existing dECM to include tissue-specific ECM [258], or the modification of the surface of synthetic substrates [84,204,205,259,260,261]. Such a technique can be used in more complex situations, such as the surgical implantation of a scaffold for autologous cells to produce the ECM, followed by explanting then decellularizing the resulting scaffold, and finally implanting it in the orthotopic location [262]. 

### 3.4. Advances in ECM Decellularization

While the above methods have been widely used in ECM decellularization, researchers have been continuously developing new ways or techniques to improve and enhance decellularization by reducing the damage to the ECM, retaining more bioactive compounds, reducing the toxic residuals, and speeding up the decellularization process (Table 3).

#### 3.4.1. Vacuum

In choosing the parameters, selection of the pressure of the vacuum is important. Friedrich E et al. (2017) used a vacuum pressure of 508 mmHg (67.7 kPa), in combination with a chemical batch of 0.25% SDS that was agitated at 90–120 rpm, and found that the anecdotally reported decellularization was faster and more efficient [146], while Wu Young M et al. (2020) used similar protocols and achieved effective decellularization [118]. Butler CR et al. (2017) compared vacuum-assisted decellularization with agitation (and lower concentrations of detergents) of the trachea with agitation in a bath of sodium deoxycholate bath under atmospheric pressure and found that, though there was no significant difference in terms of retained bioactive compounds and cell debris between the two methods: the vacuum was more effective at removing cellular material with a substantial reduction in the amount of time to achieve decellularization; there was no difference in mechanical properties of the ECM; cell morphologies in cell culture with the ECM were suggestive of more mature cellular adhesion to the ECM and higher cell survival *in vitro*; and there were no significant difference in angiogenesis, biocompatibility, and cellular integration *in vivo* [180]. Lange P et al. (2015) conducted decellularization of porcine and human trachea using a vacuum of < 1 kPa, agitation, and a combination of Triton X-100, SDS, and DNase and found that the vacuum was significantly better at removing DNA and qualitatively better at removing MHC-1 and caused no significant difference in collagen content, GAG, and biomechanical properties [263]; however, a subsequent study (2016) with rabbit trachea found that while the vacuum was effective at removing the cell contents, the vacuum damaged the ECM [183].

#### 3.4.2. Hydrostatic Washing

As a juxtaposition to the use of vacuum pressure to decellularize tissues, increasing the external pressure, known as high hydrostatic pressure, on tissues during decellularization has been shown to be effective, especially in dense tissues. Xu K et al. (2017) developed a hydrostatic washing system in which they subjected enthesis tissue to agitation in a bath of SDS and Triton X-100 at 26.7 kPa. They found that the hydrostatic washing for 24 h was as effective in removing cells and DNA as detergent baths under atmospheric pressure after 72 h [264]. Nakamura N et al. (2019) used hydrostatic treatment to decellularize rat periodontal ligament at 490 MPa at 10 °C for 10 min, followed by treatment with DNase and MgCl_2_ in saline at 37 °C for four weeks, washing with 80% ethanol in saline at 37 °C for three days, and washing in pure saline at 37 °C for another three days. They reported that the implanted decellularized periodontal ligament was able to attract host cells and partly regenerate the periodontal ligament in a decellularized mandible bone [265]. Kobayashi M et al. (2020) decellularized porcine aorta using similar protocols to Nakamura N et al. (2019), with saline baths at 1000 MPa at 30 °C for 10 min, followed by treatment with DNase and MgCl_2_ in saline at 4 °C for seven days, washing with 80% ethanol in saline at 4 °C for three days, and washing in saline at 4 °C for another three days; and they reported that hydrostatic pressure was successful in removing DNA, and the dECM had minimal histological changes compared to the considerable changes from washing with SDS [266]. In another study, Kobayashi M et al. (2020) used the same hydrostatic pressure treatment with the addition of a 600 MPa treatment to compare to the 1000 MPa treatment and the addition of citric acid in saline to the final wash to decellularize porcine SIS and UBM. They reported that, compared to sodium deoxycholate decellularization, the hydrostatic treatment had less effect on gelation potential of digested dECM, had less negative effect on the elastic modulus, and resulted in the formation of a capillary network structure by seeded primary rat brain microvascular endothelial cells [267]. Kobayashi M et al. (2022) decellularized small intestine submucosa (SIS), urinary bladder matrix (UBM), and liver fragments using the same protocol as Kobayashi M et al. (2020), though without citric acid. This technique was successful in removing cells in SIS and UBM, but was not as effective in liver tissue due to the thickness of the pieces. The authors were able to successfully collect MBVs from the dECM [268]. Charoensombut N et al. (2022) used hydrostatic pressure, increased by 65.3 MPa/min to 980 MPa, held for 10 min, and reduced by 65.3 MPa/min until atmospheric pressure was reached, to decellularize rat uterine tissue in PBS at 30 °C; this was followed by washing with 0.9% NaCl, 0.05 M magnesium chloride hexahydrate, 0.2 mg/mL DNase I, and 1% penicillin and streptomycin for seven days. The most effective washing method was to use internal radial perfusion in a bioreactor, removing 90% of the DNA content [269]. Kurokawa S et al. (2021) decellularized bovine dorsalis pedis arteries in saline hydrostatically pressurized to 1000 MPa at 30 °C for 10 min, using the same wash protocols as Kobayashi M et al. (2022), except for replacing the final wash with storage in a citric acid buffer at 4 °C. The authors were able to sufficiently remove DNA, and they reported no significant changes in the structural properties of the dECM. *In vivo*, the decellularized vascular graft lumens were recellularized with vascular remodelling after 4 weeks, though there were moderate stenoses of the grafts and no aneurysmal changes noted [270]. 

#### 3.4.3. Pulsatile Perfusion

Pulsatile flow is important to ensure proper tissue perfusion *in natura*, and it can be beneficial for perfusion decellularization. Park SM et al. (2018) compared pulsatile flow to rat hearts, and found that pulsatile flow leads to improved flow of decellularizing fluid through the vasculature, more profound decellularization, significantly lower DNA content, and no difference in collagen and GAG [165]. Strucker B et al. (2014) compared the decellularization of rat livers via the portal vein or hepatic artery, with or without oscillating external pressures to mimic *in natura* pressure changes due to respiration, and found that the oscillating pressure resulted in a more homogeneous decellularization on gross examination, lower residual DNA content, and a slightly lower alteration of ECM composition [245].

#### 3.4.4. Chemical–Penetration Enhancement

Detergents are damaging to the ECM, so it is important to find ways to improve the efficiency of the detergent during perfusion or decrease the perfusion time. Guler S et al. (2018) used the solvent dimethyl sulfoxide (DMSO), commonly used for the cryopreservation of tissues and to enhance the cellular penetration of various bioactive molecules and histology dyes, to enhance the penetration and effectiveness of SDS in the decellularization of aortic tissue in an agitated bath. They found that adding DMSO enhanced the decellularization, reduced the time needed for decellularization, reduced the amount of DNA, and better preserved and protected GAG, elastin, and collagen, and there was no significant effect on biocompatibility or cell viability and activity [166].

#### 3.4.5. Sonication

Sonication is a common method for cleaning objects and disrupting cell membranes due to cavitation, which has shown potential in decellularization of tissues. Azhim A et al. (2014) used sonication to aid in the decellularization of porcine aorta with SDS. Their results showed that sonication had little effect on the fibrous structure of ECM, was more effective at removing cells than agitation alone, and greatly reduced the decellularization time, but it decreased the pH of the decellularizing solution over time (likely the result of the sonochemical formation of acids from the dissolved gases that were induced by the sonication treatment) [216]. Suss PH et al. (2021) evaluated the use to sonication to enhance the decellularization of peripheral nerve allografts and found that sonication slightly increased the effectiveness of decellularization (non-significantly), with no additional impact to the ECM [217]. However, Tchoukalova YD (2017) found that sonication with deoxycholate and NaCl did not improve the effectiveness of removing chondrocytes from tracheas, significantly decreased the GAG content, and appeared to disrupt the submucosal connective tissue [271].

#### 3.4.6. Nonthermal Irreversible Electroporation (NTIRE)

Irreversible electroporation is a technique that uses electricity to perforate cell membranes to cause irreversible damage to cells. Electroporation has found clinical use in treating tumours by causing irreversible damage to the cell membrane of cells in the target area, with or without pre-heating by radiofrequency ablation, and has also found use in decellularization research [272,273,274,275,276]. In a preliminary study, Sano MB et al. (2018) used NTIRE to decellularize porcine livers, finding that the treatment could create sizable lesions that did not affect the ECM structure, while causing tissue disruption, cell delamination, and cell death. This study was not an attempt to fully decellularize the livers, but it does show the potential of this method as a part of decellularization [165].

#### 3.4.7. Decellularizing Agents

The chemicals used in decellularization are vital to effective decellularization, but those chemicals can have unintended effects on the resulting dECM. Finding new decellularizing chemicals is an important area of research. 

The ionic detergent potassium laurate has shown promise as a superior detergent to SDS. Obata T et al. (2019) showed that in comparison to SDS, potassium laurate is also effective at removing cells and DNA, shows better retention of ECM compounds and better-preserved architecture, shows increased cell viability and proliferation *in vitro*, and results in a lower inflammatory response and better cell distribution *in vivo*. These findings suggest that potassium laurate might be a good detergent for use in decellularization [190].

Sodium lauryl ether sulfate (SLES) is another ionic detergent that is being used in tissue engineering. Kawasaki T et al. (2015) conducted a study comparing SDS and SLES for perfusion decellularization. Much of the dECM content assessed was similarly preserved with either detergent, though SLES had better GAG retention and better-preserved microarchitecture and had a significantly lower inflammatory response and platelet adhesion when the dECM was tested *in vivo*. However, the authors also found that SLES was slower at decellularizing that SDS [149]. Liu G et al. (2018) used SLES in an agitated bath to decellularize breast cancer tissue and found that it was effective at removing cells, but that higher concentrations and prolonged exposure times could lead to damage to the ECM and structural proteins and lead to increased losses of proteins and GAG [127]. Naeem EM et al. (2019) decellularized murine livers by perfusion to compare SLES with the commonly used detergent combination of SDS and Triton X-100. They found that while decellularization with SLES was slower that SDS/Triton X-100, the microvasculature structures were better preserved, and the dECM retained much more GAG and collagen I and IV, though there was no difference in the clearance of DNA. There was a considerable difference when the decellularized livers were used *in vivo* to repair a small, surgically induced lesion in a native liver. Liver dECM treated with SDS/Triton X-100 resulted in the death of all of the animals, whereas there were no deaths in the animals treated with liver dECM decellularized using SLES. The SLES–treated liver dECM grafts were repopulated by native cells and the architecture was regenerated *in vivo*. When used to replace an entire lobe, the SLES-treated dECM was found to also have architecture that regenerated the liver, and a large population of cells had migrated into the dECM and performed the appropriate physiological functions [188]. These findings suggest that, with appropriate use, SLES might be a good detergent in decellularizing tissues.

Most decellularization techniques involve the use of detergents as the decellularizing agent, but other chemicals, such as NaOH, can be used to decellularize tissues. Sengyoku H et al. (2018) compared the decellularization of murine lungs, using NaOH (pH 12) with both CHAPS and SDS. They found that NaOH could remove cell components as effectively as the detergents; collagen, GAG, and adhesion protein retention were similar to the detergents; and there was a significant reduction in DNA content with NaOH. *In vitro*, the lungs treated with NaOH and SDS were equally populated with functional cells. This presents another alternative to the standard detergents [208].

#### 3.4.8. Supercritical Fluids

Supercritical fluids, such as CO_2_, achieve a strange state where they act both as a gas and a liquid. This presents some interesting opportunities for using supercritical fluids to enhance or replace common decellularization chemicals. Often, a co-solvent, such as ethanol, is used in combination with the supercritical CO_2_ (scCO_2_), and the co-solvent dissolves into and saturates the supercritical fluid [232]. A small amount of water is sometimes added to reduce the dehydration of the dECM. Seo Y et al. (2018) developed a decellularization technique using scCO_2_ with ethanol as a co-solvent to decellularize a heart. They lyophilized murine heart tissue, cut it into small pieces, then soaked it in 100% ethanol in a scCO_2_ chamber at 350 bar for six hours, and compared the results with decellularization using a combination of SDS and Triton X-100. They found that scCO_2_ plus ethanol was comparable to the detergents in cell and DNA removal and was superior in the retention of GAG, soluble collagen, adhesion proteins, and angiogenic factors. However, due to the increased retention of ECM compounds, the solubilized dECM created from the scCO_2_ plus ethanol dECM needed to have additional collagen added for gelation to occur. *In vivo*, the solubilized dECM plus collagen showed more signs of neovascularization to the detergents but no difference in immune response [201].

#### 3.4.9. Alternating Decellularizing Solutions

One of the goals of finding new ways to decellularize tissues is to find ways to limit the exposure time of harsh detergents. One method that was developed by Poornejad N et al. (2016) was to use repeating cycles of alternating solutions of hypertonic NaCl, SDS, and deionized water (hypotonic), with increasing flow rates and a constant pressure, to decellularize porcine kidneys. This causes a large osmotic gradient to form during the decellularization and, compared to a standard decellularization protocol with SDS, resulted in a slower decellularization process (approximately twice as long), comparable DNA removal, significantly higher preservation of GAG and soluble collagen, and similar levels of growth factors. *In vitro*, there was no difference in cell attachment, but there were higher cell viability and improved gene expression patterns [228].

## 4. ECM Modification and Methods

The ECM has many beneficial properties for tissue engineering, but it also has some limitations. Decellularized ECM, especially solubilized dECM hydrogels, has reduced mechanical strength and can degrade *in vivo* too rapidly to allow for sufficient tissue repair and regeneration, though this behaviour differs based on the original tissue source, suggesting that there is a need to stabilize the dECM to more effectively promote repair and for use in tissue engineering [163,170].

### 4.1. Improving Structural Stability

Improving the stability of dECM can be achieved by adding materials it. This can include other materials that are incorporated directly into the dECM that provide structural strength, and materials that are placed around the dECM to provide structural supports to reduce the forces being exerted on the it. The stability can also be improved by cross-linking the dECM to stabilize the proteins.

#### 4.1.1. Composite dECM Scaffolds

There are different ways to make hybrid scaffolds. This can include modifying intact dECM, modifying solubilized dECM with hydrogel additives, and adding external structural supports. 

Solubilized dECM can be modified through the addition of other hydrogels. One common hydrogel that is combined with dECM is sodium alginate. Sodium alginate, commonly referred to as simply alginate, is favourable in that it is easily and rapidly cross-linked using a solution with calcium ions, such as CaCl_2_. Combining alginate with solubilized dECM allows the desired scaffold to rapidly stabilize, giving time for the dECM to gel [97,203,277,278]. If desired, the alginate can be removed later by adding sodium citrate or EDTA to bind with and remove the calcium ions stabilizing the alginate. Techniques such as this one allow for the encapsulation of dECM and cells in microspheres and in 3D-printed/plotted strands to create scaffolds (e.g., plotting into a CaCl_2_ solution, using coaxial printing by having a core with a source of calcium ions and a shell with dECM mixed with alginate) [14,160,182]. Alginate or gelatin can also be used to encapsulate dECM in powder form in denser scaffolds [213].

Chitosan is a polysaccharide derived from the chitin that makes up crustacean shells. Chitosan can be solubilized in an acid and then mixed with solubilized dECM or added to intact dECM. Adding chitosan to solubilized dECM to create scaffolds can be used to add antimicrobial properties and to modify the mechanical properties and pore sizes of the scaffold [110,115,197,279,280]. Chitosan can also be added to dECM to allow for photo-cross-linking to tissues with the right cross-linking agent [281].

Sheets of dECM can be modified with additional dECM to better target specific tissues. Using simple tissues such as amnion, blood vessels, and the small intestine, they can be modified to use as a substrate by adding solubilized dECM from another tissue or by culturing specific cell types on the ECM to lay down new extracellular matrix [111,198,258].

Collagen and its derivatives can be used to support solubilized dECM. The self-assembly of collagen is one of the main initiators of the sol–gel transition of solubilized dECM, and adding solubilized collagen can help to improve the strength of the gel [201]. Gelatin, a hydrolysed form of collagen, can be mixed with solubilized dECM as a carrier, but gelatin melts at 37 °C, while ECM gels at 37 °C, so careful planning is necessary [97,115,139,223]. Gelatin methacryloyl (GelMA) is a popular hydrogel used in tissue engineering, due to having similar biocompatibility and mechanical properties as gelatin, but the benefit is that GelMA can be easily cross-linked with the addition of a photoinitiator [237].

For scaffolds requiring more strength, dECM is often combined with stiffer materials. For biological materials, decellularized bone can be used as a structural support, especially for cartilage and bone scaffolds [236]. Synthetic plastics are also widely used. They can be used as a coating on spherical dECM hydrogel scaffolds [14], mixed with solubilized dECM to improve the mechanical properties [168,282], printed as a structural support around solubilized dECM [82,136,162,213,283,284], or electrospun to support intact dECM [285]. Bioactive compounds added to the dECM can influence the biological response to dECM materials. Liver dECM treated with prednisolone in combination with cell seeding has been shown to reduce the presence of inflammatory immune cells in rats, while also significantly increasing the *in vivo* regeneration and angiogenesis response [189]. Curcumin has been shown to have antimicrobial effects in dECM scaffolds and result in high cellular metabolic activity with no apparent impact on cell viability [286]. Adding exosomes with a collagen-binding domain peptides to more effectively secure them to the dECM has the potential to enhance cell proliferation *in vitro* and reduce fibrosis *in vivo* [253]. The addition of exosomes collected from media cultured with bone marrow–derived mesenchymal stem cells to dECM bioinks have been shown to restore chondrocyte mitochondrial dysfunction, improve chondrocyte migration, and enhance the polarization of macrophage populations towards an M2 phenotype [287].

When using 3D-printing techniques to create a scaffold using solubilized dECM, there are times where sacrificial materials are needed in order to create internal structures. A hydrophilic surfactant, Pluronic F-127, has favourable properties for use as a sacrificial material with solubilized dECM: at concentrations about 20%, Pluronic F-127 can form a gel and remains liquid at low temperatures but solidifies at ambient temperatures [288,289,290]. These properties are suitable for using with solubilized dECM, as solubilized dECM gels at 37 °C but remains liquid below that, while Pluronic F-127 is solid at 37 °C but can be solubilized by decreasing the temperature below the gel point or by increasing the hydration. This allows Pluronic F-127 to be used as a sacrificial gel for creating moulds or as the core for the coaxial printing of solubilized dECM to make a hollow tube (e.g., a vascular scaffold) [160,171]. Other novel additives include Laponite-XLG nanoclay, poly(ethylene glycol)-diacrylate, and poly(*N*-isopropylacrylamide) (PNIPAM) [291,292]. The materials can enhance the printability and structural stability of the scaffolds.

#### 4.1.2. Cross-Linking

Cross-linking is a popular option for stabilizing hydrogels of all sorts. Solubilized dECM has the ability to undergo autogelation, but there are situations where chemical cross-linkers are used. The two main types of exogenous cross-linking agents are chemical cross-linking agents and photoinitiators. Cross-linkers, directly or through coupled reaction, induce proteins to cross-link. In the case of dECM, common cross-linking agents include glutaraldehyde [110], glyoxal [223], genipin [293], and *N*-hydroxysuccinimide (NHS), often in combination with 1-ethyl-3-(3-dimethylaminopropyl)-carbodiimide (EDC) [211,225,236] to stabilize the reaction, among other chemical cross-linking compounds, to chemically cross-link proteins to one another. Photoinitiators also induce protein cross-linking, but their mechanism is to generate reactive intermediates that initiate the cross-linking. Many photoinitiators, such as Irgacure 2959 [139,168], riboflavin/vitamin B_2_ [136,161,169], and rose bengal [281] require excitation with UV to initiate the cross-linking process. Since UV can be harmful to cells embedded in a scaffold, other photoinitiators that work with visible light, such as Eosin Y [237], are being used to cross-link solubilized dECM. A novel cross-linking method, developed by Nishiguchi and Taguchi (2021), involves tethering genipin to four-arm PEG using pH-driven processes to create an adhesive that induces a rapid cross-linking reaction and as a gelation agent [294]. These methods allow both solubilized and intact dECM to be cross-linked.

#### 4.1.3. Structural Fabrication

Cell-cultured dECM sheets can be modified to create scaffolds. Magnan et al. (2020) cultured human skin fibroblasts to create sheets of dECM that they were able to be modified to create yarns. These yarns were made by cutting the sheets of dECM into strips and then twisting the strips to form threads. These threads could then be twisted to spin yarns or braided to form stronger multifilament strands. These yarns could be weaved, knitted, knotted, or used as suture threads. For creating vascular grafts, the yarns could be woven on a circular loom to create a mesh with a warp and weft structure to produce a very dense wall to the graft with ends that were suitable for suturing and low permeability [257].

Physical cross-linking of hydrogels is another widely used process. For dECM, the two most common methods both use vacuum. For sheets of dECM, these can be stacked together and compressed using vacuum pressure to laminate the dECM sheets together [225,226]. Lyophilizing hydrogels typically results in a physically cross-linked porous sponge. Lyophilized dECM hydrogels create a highly porous structure, typically with interconnected pores (though the freezing process prior to lyophilizing can affect this), and are hydrophilic [110,194,196,197,199].

Lyophilized hydrogel sponges usually allow for improved cell migration and flow of nutrients into the interiors of the scaffold, depending on the pore structure. Another method to create composite, multilayer scaffolds is to fabricate an interlocking hole-and-tab system to interlock sheets of biomaterials, without the need to add cross-linking chemicals [295].

### 4.2. Improving Fibrous Structure 

*In natura*, the ECM is, macroscopically, a fibrous structure. The fibrous structure provides structural supports, and the arrangement of the fibres leads to different physical properties. There are different ways to mimic this fibrous structure to allow for improved integration of the dECM *in vivo*.

#### 4.2.1. Electrospinning Solubilized dECM 

In combination with carriers or as an additive to synthetic materials, dECM can be electrospun to form nanoscale fibres in various arrangements that can be tuned for the desired use. Conventional electrospinning is carried out using DC power, however, AC electrospinning is possible, with the advantage of not needing a grounded collector plate [296,297,298]. The use of a needle as an electrified spinneret is common in electrospinning; however, the rate of production with a single needle is slow, whereas needleless electr numerous fibres simultaneously [299]. Directional fibres can be electrospun with high voltage by collecting them on a rastering, rotating mandrel rotating at high speed [138,300,301,302,303] or on parallel grounded electrodes [304,305] and at low voltage by using initiators on an X–Y translational stage moving linearly back and forth [223,306]. Perpendicular fibres can be created by using paired parallel electrodes at 90° to one another in a cross arrangement and alternating which parallel pair of electrodes are grounded [305]. Radial aligned fibres can be collected using a cup-and-pin arrangement, where the centrally located pin and the circumferential edge are grounded [305]. Random mesh patterns can be fabricated by using high voltages to trigger random movements of the generated fibres before they are collected on a stationary plate [301,302,303,305,307,308]. Tubular scaffolds can be fabricated on a rastering, rotating mandrel and are slid off of the mandrel to retain their shape, as opposed to being cut off the mandrel to form a flat sheet [304]. Coaxial electrospinning, also known as core-shell electrospinning, can be created using specially designed spinnerets, to allow for simultaneous dispensing of fluids in the desired arrangement, and varying the flow rates will affect the thickness/diameter of the different parts of the fibre [309]. Wet electrospinning relies on a grounded bath as the collector so that the fibres do not dry out or to induce coagulation of the fibres [310]. Using a more complex layout, yarns can be spun by directing the electrospinning to a grounded rotating plate or funnel, then drawn and spooled onto a rotating mandril [311]. There are variations on all of these techniques; see Figure 2.

Producing electrospun dECM generally requires the use of carrier agents. These carrier agents allow for lower electroconductive and physical properties needed for electrospinning as well as solubilization of dECM if the dECM has not been enzymatically digested. Due to the high surface tension of water and the limited solubility of many proteins, dECM alone, like other natural polymers, does not electrospin very well [312], but dECM can form microscopic beads through electrospraying [138,312,313]. By solubilizing dECM in a combination of acetic acid and ethyl acetate or HFIP (1,1,1,3,3,3-hexafluoro-2-propanol), the surface tension of the solution is reduced, and the proteins are denatured, allowing for the electrospinning of fibres [223,300,301,302,303,305,307,308,312]. Another option is to solubilize the dECM using a solution of acetic acid:ethyl acetate:ddH_2_O (*v*/*v*/*v*) = 3:2:1) at 45 °C for three days, then adding an equal mass of synthetic polymers, such poly(ε-caprolactone) (PCL), and allowed to mix for two days [314]. Blending dissolved dECM solution with PCL allows for the significant improvement of the cell viability, function, and proliferation in a concentration-dependent manner compared to the polymer alone [303,305,307,308]. These techniques can allow for the customization of dECM scaffolds in terms of physical, mechanical, and bioactive properties.

Xia B et al. (2020) combined powder meniscus dECM with PCL for electrospinning, and they reported that dECM improved the hydrophilicity of the scaffold, enhanced cell proliferations and spreading, and upregulated fibrochondrogenic gene expression [314]. Li H et al. (2021) fabricated twisted yarns from the electrospun fibres of digested meniscus dECM and poly(lactide-co-caprolactone), at a 1:9 ratio and solubilized in HFIP, and they reported that the twisted fibres resulted in the improved expression of meniscus-associated genes and production of dECM, considerable cell infiltration, and faster remodelling and degradation of the scaffold in vivo compared to randomly arranged fibres. Reid JA and Callanan A (2019) used a low concentration of aortic and cardiac dECM mixed in PCL (~3:97) dissolved in HFIP, which was spun and collected on a rotating mandrel. Even at this low concentration of dECM, human umbilical venous endothelial cells (HUVECs) reached 100% confluence in 10 days on the aortic dECM-containing fibres, compared to 50% for cardiac dECM-containing fibres and 30% for PCL, suggesting that vascular ECM is important for the viability and proliferation of HUVECs. They also reported that the inclusion of dECM increased the Young’s modulus and hydrophilicity of PCL [315]. Sobreiro-Almeida R, Fonseca DR, and Neves NM (2019) combined kidney dECM with PCL at ratios of 70:30, 50:50, and 30:70, respectively, dissolved in HFIP for electrospinning on a flat–plate collector, and cross-linked with glutaraldehyde vapours after electrospinning. They reported a concentration–dependent decrease in Young’s modulus under compression as the concentration of dECM increased. They reported that the addition of dECM significantly improved the cell proliferation of kidney tubule cells and that higher concentrations of dECM resulted in better cell distribution and cell spreading over the scaffold and improved formation of cell–cell tight junctions [308]. Patel KH et al. (2019) prepared 10% PCL, 5% dECM:5% PCL, and 10% dECM from muscle dECM, dissolved in HIFP, and collected on parallel fibres on a rapidly spinning disc-shaped mandrel, which was modified to enhance the fibre alignment by increasing the electric field intensity. They reported an increase in the Young’s modulus under tension with the addition of dECM, and the 10% dECM fibres were removed from the study due to their poor mechanical properties (the dECM fibres also required cross-linking to maintain structural stability when hydrated). They also reported that the aligned PCL:dECM fibres resulted in more myoblast proliferation compared to the PCL fibres [303].

#### 4.2.2. Modifying dECM for Enhanced Engraftment

Intact dECM has been shown to work well for many purposes, but there are ways to improve the engraftment of dECM *in vivo*. Tendons and ligaments, for example, attach to the bone in a fibrous structure called the enthesis. If there is a need to replace a ligament, often a plug of bone is taken with the ligament attached to improve the engraftment. Liu H et al. (2017) modified sections of decellularized tendons by treating both ends of the tendon in acetic acid followed by sonication in order to disorganize the dECM fibres at the ends of the tendons. This created a random-aligned-random architecture of the tendon dECM. This technique was used to replace the excised native ACL with the modified tendon in rabbits, and the modified tendon was significantly better at bonding the adjacent tissue, inducing bone formation, and forming fibrocartilage than unmodified tendons [215].

### 4.3. In Vivo Use of dECM

The idea of using dECM for tissue engineering has long history. Work going on in the Badylak Lab at the University of Pittsburgh McGowan Institute for Regenerative Medicine has focused extensively on the use of dECM as a scaffold material for tissue engineering for decades. Beginning back in 1987, when conducting experiments on canines, Stephen Badylak experimented by replacing a section of the aorta with a section of the small intestine (Badylak admits that this was an “outside-the-box experiment that probably never get past a university animal-care committee today”), with astounding results that almost no one could believe—the small piece of intestine became an artery, even histologically. Switching from a canine small intestine, Badylak repeated the experiments with both feline and porcine dECM, expecting to see an immune response in the larger studies on canines, but the results were the same as the previous experiments. In 1989, Badylak tried another radical experiment: removing a section of canine Achilles tendon and replacing it with porcine dECM, and the tendons completely regrew with no scarring or loss of function. Through this continued work, Badylak learned that the body degrades the implanted dECM and uses the material to grow new, healthy tissue. The potential for dECM in *in vivo* tissue engineering was there [316,317,318,319].

Extracellular matrix has been shown to work well *in vivo*. As previously discussed, not only does dECM induce minimal inflammatory immune response, but host cells are also able to migrate into the dECM and begin to grow new tissues. The use of dECM *in vivo* leads to increased vascularization of the implants [91,156,157], faster healing and regeneration [92], and significantly improved functional recovery in spinal cord injuries [57]. Adding stem cells to dECM has been shown to improve the repair of growth plates [194], maintain more patent vasculature with less thrombus formation [152], and provide evidence of tooth regeneration [156] when compared to acellular dECM. Successful use of dECM *in vivo* has led to a great deal of interest in the clinical use of dECM for treating diseases.

### 4.4. Solubilizing dECM for Bioinks

The dECM that is derived from decellularizing simple tissues in chemical baths or more complex tissues through perfusion can be used for conducting experiments, but another popular option is to produce a hydrogel, by solubilizing dECM through enzymatic digestion. The solubilized dECM colloidal dispersion continues to demonstrate the desirable bioactive properties of the intact dECM, and the biological response is often superior to that of a single protein-based cell culture [9,10,11]. If using more than one tissue source for the dECM, such as a combination of cartilage and bone, complex 3D scaffolds can be fabricated with specific dECM placement, to better mimic *in natura* anatomical arrangements [320]. With the ability to customize the shape and placement of dECM by rapid prototyping methods, this area of research is growing.

Solubilizing dECM is a multistep process. The first step of solubilizing dECM is to turn it into a powder. There are various methods for doing this, such as freezing with liquid nitrogen and using a mortar and pestle [82,91,125,136,159,161,212], milling with a rotary or cryogenic mill [20,23,87,110,138,139,168,171,175,201,223,237,279,320,321], or making a slurry with a homogonizer, followed by lyophilizing [302]. A variation on this process involves saturating the dECM with NaCl, snap freezing the dECM to precipitate the NaCl crystals and separate the protein fibres, and then milling it into a powder, followed by multiple washes to remove the NaCl, centrifugation, snap freezing of the pellet, and lyophilizing and disaggregating the powder; however, this process is not often used for preparing dECM for solubilizing [321,322]. Once the dECM has turned into a powder of suitable particle size, the powder is lyophilized (if not done before milling), and then it can either be used directly as an additive [102,175,279,280,323] or be digested to create a liquid dECM mixture.

The digestion process for dECM is typically a combination of an acid with pepsin. The most common acid used is HCl at a concentration of 0.01 M [14,56,82,86,87,90,92,109,111,117,138,143,163,168,171,172,173,187,195,197,199,200,202,203,211,227,237,321,324], though acetic acid at a concentration of 0.5 M is another popular acid [83,96,107,114,115,125,136,139,159,160,161,162,201,320], with both used at a ratio of 1 g of dECM powder:100 mg pepsin:100 mL acid and stirred for 48 h. There are different effects of the type and concentration of acid used, with 0.1 M HCl accelerating the digestion, creating higher cell viability, migration, and proliferation, and enhancing the ability to induce tissue formation, though it results in a softer and more unstable gel than acetic acid [325]. There are also optional steps after solubilizing dECM; the first is to filter the colloidal dispersion to remove the undigested particles that might affect later use [125,159,161,210], and the second is to use NaCl to precipitate the dECM [139]. After digestion, the pepsin can be deactivated by raising the pH to physiological conditions through the use of PBS and NaOH, generally at a ratio of 10× PBS and 0.1 M NaOH in proportions of 1:10 and 1:9, respectively, of the volume of the digestion mixture. Extracellular matrix bioinks will remain liquid at cold temperatures and undergo a sol–gel transition under physiological conditions (i.e., pH 7.4 at 37 °C), after approximately 15–30 min [14,114,162,163,200,202]. The degree of digestion of the dECM affects the printability and stability of the scaffold in 3D printing, with a less digested dECM resulting in a higher shape fidelity of the scaffold structure, an improved stacking accuracy of the extruded strands to increase the number of layers, and a better structural stability of the scaffold [326].

#### Inks and Bioinks

Creating inks and bioinks from hydrogels is a popular method of incorporating cells into a scaffold created by using 3D printing and other additive manufacturing techniques. A hydrogel that does not contain cells is considered an ink, whereas a hydrogel with incorporated cells is a bioink. One very beneficial feature of solubilized dECM is that the colloidal dispersion is shear-thinning. This means that when solubilized dECM is being dispensed through a nozzle, the shear stress that would otherwise damage cells is reduced (using a tapered print nozzle helps to reduce shear-induced damage to cells as well) [327,328,329,330,331,332]. This shear-thinning behaviour helps to cushion cells during dispensing, and, after dispensing, the dECM hydrogel returns to a higher viscosity that holds cells in place rather than allowing them to settle out [82,114,333]. The printability and ultimate structural stability of dECM colloidal dispersions will depend on the viscosity of the suspension following dispensing, as well as how rapidly the dECM can gel to stabilize the strands, which might necessitate the use of a plotting bath [334,335].

There are several different additive manufacturing techniques that are often collectively lumped together under the umbrella of 3D printing, but there are important differences. The technique that most people are familiar with in 3D printing is a technique called fused deposition modelling, in which a fluid is extruded from a nozzle translating in the x and y directions, moving in the z direction with every new layer. Another closely related technique is 3D plotting, in which the ink is dispensed into a liquid bath that provides buoyancy to support the scaffold and can have additives that induce cross-linking (e.g., alginate dispensed into a bath of CaCl_2_) [336,337,338]. When simultaneously using different dispensing heads, this can allow for the precise placement of tissue-specific dECM to create a complex 3D scaffold [212]. When dispensing cell-laden bioinks, these processes are referred to as 3D bioprinting and 3D bioplotting, respectively.

Using 3D-printing techniques does allow for specialized scaffolds to be created. Using coaxial printing techniques, Gao G et al. (2017) were able to fabricate scaffolds for small blood vessels. To create the shell of the strand, they combined dECM with alginate, and the core was Pluronic F-127 to which CaCl_2_ had been added. As the strand was extruded, the calcium ions diffused from the Pluronic F-127 to cross-link the alginate in the shell, thereby stabilizing the dECM for gelation, and the Pluronic F-127 provided structural supports as a gel to keep the printed strand from collapsing. The Pluronic F-127 was easily removed later, leaving behind a small, tube-shaped scaffold for blood vessels [160]. Kim MK et al. (2020) developed a bioink using powdered liver dECM in a mixture of 3.75% gelatin, 0.3% hyaluronic acid, and 0.3% fibrinogen and compared this with a dECM bioink at the same concentrations of dECM but with no additives. They found improved printability for the bioink fabricated from dECM powder (though they did not use a plotting bath) [339]. Lee J et al. (2020) developed an ink using methacrylated demineralized bone dECM with 2.54% alginate, 0.15% CaCl_2_, and 0.3% Irgacure that could be cross-linked through a combination of UV treatment and aerosolized CaCl_2_. They found that bioinks with a concentration greater than 3 wt% dECM had high shear stress during dispensing that significantly decreased cell viability and that their 2% dECM composite bioink had similar printability to 3.5% alginate [340].

Another additive manufacturing process that is showing promise in tissue engineering is stereolithography, and dECM inks are suitable for this technique. Mao Q et al. (2020) used digital light processing (DLP) bioprinting to fabricate a complex, internal ring gear-like scaffold, using a bioink made from liver dECM and GelMA. They found that the addition of liver dECM to GelMA improved the printability and cell viability of GelMA [341]. Elomaa L et al. (2020) used DLP to successfully fabricate multilayer scaffolds from methacryloyl-functionalized liver dECM and star-shaped poly(ε-caprolactone) methacrylate [342]. Wang D et al. (2022) used DLP to fabricate a honeycomb-shaped scaffold from pancreas dECM and hyaluronic acid methacrylate, which, compared to hyaluronic acid methacrylate alone, allowed for the development of high-density vascularization, superior cell viability, and the appropriate regulation of blood glucose levels [343]. Chen P et al. (2019) used DLP to fabricate a radially oriented scaffold for treating cartilage defects, and they reported that scaffolds made with dECM and GelMA (with or without exosomes) showed superior repair and regeneration of articular cartilage than GelMA alone [287].

The use of additive manufacturing processes such as 3D printing allows for highly customizable scaffolds. Behre A et al. (2022) used a CT scan to model a volumetric muscle loss injury in canines and generated a 3D-plotted patch for filling the wound using a 1:1 mixture of digested urinary bladder matrix and collagen dispensed into a gelatin microparticle plotting bath (freeform reversible embedding of suspended hydrogels). The patch was able to make better contact with fewer air voids between the patch and the tissue than an onlay EMC sheet (99.9% touching the wound compared to 90.3%, respectively). They also demonstrated that it might be possible to create similar scaffolds for volumetric muscle loss injuries in human patients [344]. Chen L et al. (2022) developed a hydrogel from dermis dECM that they used as a 3D-printed, glutaraldehyde-cross-linked underlay for split-thickness skin grafts. The dermal analogue grafts reduced wound contraction and the formation of scars, with a higher population of polarized macrophages towards the M2 phenotype and reduced inflammation, compared to grafts without the dermal dECM underlay [345]. Kim BS et al. (2018) used a combination of 3D printing and inkjet printing to fabricate a full-thickness skin model, consisting of dermis and epidermis layers, from porcine skin dECM. They reported that, *in vitro*, the scaffold stabilized with minimal shrinking, and, *in vivo*, there was accelerated wound closure, epithelialization, and vascularization of the scaffold [346].

### 4.5. Sterilization

Sterilization of the dECM is important for both *in vitro* and *in vivo* use. There are a variety of different sterilization techniques used for dECM that are outlined in Table 4. The most common method used in sterilizing dECM is the use of peracetic acid and/or ethanol. Peracetic is a strong oxidizer that has effective antimicrobial properties and can be used for decellularizing tissues. Antimicrobials, such as antibiotics and ionic liquids, are also very common, and they are generally used during and after decellularization to limit the bioload during decellularization. Other, more traditional, methods of decellularization, such as UV, ethylene oxide, and γ radiation are used, but can affect the properties of the dECM. Supercritical CO_2_ shows the potential for the effective sterilization of dECM, but it can affect the mechanical properties of the dECM. Sterile filtration is useful for solutions of soluble ECM components, but it can result in the loss of proteins in high concentrations of dECM solutions and colloidal dispersions. 

### 4.6. Cell Seeding of dECM

An dECM scaffold is not of much use unless something can be accomplished with it. Though not always required for some *in vivo* uses, methods for seeding dECM for both *in vitro* and *in vivo* use are important considerations for the final experimental results. Much of the work of recellularizing dECM has made use of stem cells and immortalized cell lines (which are distinct from stem cells that in that immortalized cell lines cannot be used *in vivo* due to the uncontrolled nature of their cell growth). There are a lot of developed methods that are showing great promise.

In the last several years, there has been much discussion about stem cells. Stem cells are defined by their potential to self-renew, and there are five different types of stem cells that are defined by the degree of potency:Totipotent stem cells that can produce all of the cell types in a foetus (including the birth-associated tissues: placenta, amnion, etc., that are derived from the trophoectoderm);Pluripotent stem cells that can produce cells from all three germ layers (i.e., endoderm, mesoderm, and ectoderm);Multipotent stem cells that can produce a limited number of cell types of different lineages within one of the germ layers (though some research is showing that there is the potential for these stem cells to differentiate into the cell types of other germ layers);Oligopotent stem cells can produce two or more cell types within a specific tissue (these are sometimes divided from multipotent stem cells as an intermediate step); andUnipotent stem cells are terminally differentiated stem cells that can produce cells of only one type [351].

These cell types are derived in different ways. Totipotent stem cells are derived through cloning; pluripotent stem cells are derived from cells in the blastocyst stage of embryo development or through induction (called induced pluripotent stem cells, which is often shortened to iPSC); and multipotent stem cells can be collected from tissues (such as mesenchymal stem cells) or through induction. Another category of cells is nullipotent, which are cells that have lost their ability for self-renewal, meaning that they are terminally differentiated and do not go through cell division.

Perfusion recellularization is a popular approach. Just as perfusion decellularization is used to decellularize tissues because the vasculature reaches all parts of the tissues, seeding cells through the vasculature can help to achieve more even distribution of cells in the scaffold and less damage to the scaffold than other techniques (e.g., injection with a needle).

Following their success with perfusion decellularization of a murine heart, Ott et al. (2008) decided to try growing a working heart. Using their new heart scaffold, they injected neonatal cardiac cells into the scaffold, and perfused the vascular tree with an oxygenated culture medium. However, cells also need physiological cues and stresses to help them to mature and function properly. To achieve the desired conditions, the team provided cardiac and coronary perfusion in a pulsatile fashion in a complex circuit and provided electrical stimulation to trigger the rhythmic and coordinated contraction of the cells. After eight days, the hearts began to beat and was able to achieve 2% of an adult rat heart’s contractile force. Though histological analysis at day 8 showed only 33.8 ± 3.4% recellularization, this experiment was a major accomplishment. They also conducted separate experiments on seeding the scaffold vasculature with endothelial cells through perfusion (they achieved re-endothelialization of the coronary vessels and, simultaneously, of the ventricular cavities), as well as several other mammalian solid organs [241].

Taking on decellularizing and recellularizing of another organ, Ott HC et al. (2010) tackled lungs and went a step further with the orthotopic implantation of recellularized murine lungs. Specific cell types were seeded into the lungs via the trachea and the pulmonary veins and arteries (rat fetal lung cells and HUVECs, respectively), and the scaffold was perfused with oxygenated culture media (with 5% CO_2_) through the pulmonary arteries, started on liquid ventilation of the lungs after ensuring sufficient time for cell attachment to occur, and started ventilation with 98% FiO_2_ five days later. *In vitro*, the recellularized lungs supported cell growth that appeared similar to *in natura* states, and there were no signs of tissue necrosis, which suggested that there was intact tissue perfusion via the vasculature, and they achieved a gas exchange performance of 30% compared to fresh lungs. *In vivo*, the orthotopically transplanted lungs performed with no parenchymal bleeding, pulmonary edema, or air leakage, the lungs effectively performed gas exchange (though lower than intact lungs), and the rats were able to be extubated for several hours before they started to produce a substantial amount of nonbloody pulmonary secretions that required the rats to be reintubated. On gross examination of the lungs after the experiment, they found that the lungs were edematous and required a high inspiratory pressure. Regarding histology, the lungs were well-perfused with blood, and there were no hematomas, airway bleeding, or thrombus formation seen, but there were proteinaceous fluids in the distal airways and alveoli, in keeping with the signs of pulmonary edema [242].

This was not the end of this work. In 2013, the Ott Lab focused their efforts on another organ, the kidney, by adapting techniques from their previous work on hearts. They decellularized rat kidneys and reseeded them with epithelial and endothelial cells. Using a vacuum pressure of 40 cm H_2_O to ensure cell dispersion throughout the parenchyma, Song et al. (2013) perfused both the vascular tree through the renal artery with HUVECs and through the urine tubule structures, by using the ureter with rat neonatal kidney cells (positive pressure cell seeding was not successful, in this regard). To achieve physiological conditions, the seeded scaffolds were cultured with a whole-organ biomimetic culture in a perfusion bioreactor. Both *in vitro* and *in vivo*, the kidneys produced rudimentary urine and functioned at approximately 50%, and, *in vivo*, they demonstrated normal perfusion without signs of bleeding or clot formation and immediate graft function. The recellularized kidneys were relatively immature, but showed the potential for growing complex organs using decellularized scaffolds [243].

Pushing the boundaries of decellularizing and recellularizing scaffolds even further in 2015, the Ott Lab began working on the composite tissue regeneration of soft tissues. This time, Jank BJ et al. (2015) focused on a rat forelimb. Following the previous organ protocols, the limb was amputated and decellularized with detergents and phosphate buffer solution, though fasciotomies were performed prior to perfusion to allow the muscle areas to clear cell debris. The brachial artery was perfused with HUVECs on day 1, and the muscle areas were injected with a mixture of mouse embryonic fibroblasts, mouse myoblasts, and HUVECs on day 2, with full thickness skin grafts on day 10. The limb was perfused with an oxygenated growth medium in a bioreactor, and the started electrical stimulation on day 6. On days 14–16 of the experiment, the limbs were able to achieve approximately 80% strength of the specific force of neonatal muscles [145].

The recellularization techniques for intact organ dECM developed by the Ott Lab are still widely used. Seeding of cells though vascular perfusion, with or without perfusing additional arborized structure in the organs (e.g., airways, both arteries and veins, ureter, biliary tree, etc.) is still widely used [117,135,152,164,165,177,193,214,229], and vacuum-aided cell seeding of dECM has also been used [164]. The use of conditioning the dECM structures by mimicking *in natura* functions is also widely used.

There are other techniques for seeding cells onto dECM. For sheets, moulded gels, and some 3D-printed constructs, a common technique is to simply seed the cells directly on top of the dECM using a pipette or similar device [14,88,112,113,116,123,127,128,134,137,155,167,171,179,180,181,198,199,200,211,215,223,230,236]. For bioinks, cells are mixed into the dECM colloidal dispersion prior to gelation, providing good cell distribution within the scaffold [14,83,86,114,125,139,159,160,161,162,168,171,182,200,213,237]. Cells can be injected directly to the dECM at specific locations as well, but this can cause damage to the ECM and uneven cell populations [125]. Co-culture of different cell types to more closely resemble *in natura* cell populations in a tissue often results in synergistic effects, including improved cell viability and function, over monotype cell populations [114,136,154,171,200,223,237,249].

Another technique developed by the Ott Lab for reseeding cells is the use of a spray application. Achieving effective cell distribution and reducing cell clumping with cell seeding can be challenging, especially with some tissue types. Using a custom 3D spray system and a commercial airbrush low air pressure, Schwartz DM et al. (2017) were able to seed intestinal organoids onto pieces of decellularized jejunum. Though the cell viability of the sprayed cells was lower than pipetting to a significant amount, the increase shear force from the air disaggregated the cell clumps and organoids, allowing for improved cell distribution across the scaffold (cell viability decreased with increases in air pressure). After 7 days, this technique resulted in a continuous monolayer of cells in direct contact with the dECM, resembling the absorptive surface found on intestines *in natura* forming on the dECM, while pipetting organoids resulted in sporadic placement of cell clumps [144].

Cell seeding prior to implantation can also have beneficial effects for physiological outcomes. One problem with dECM is that it rapidly degrades *in vivo*. In a study of repairing abdominal wall defects in rats, Nowacki M et al. (2015) seeded rat mesenchymal stem cells onto dECM sheets derived from porcine aorta, seven days before implanting into an abdominal wall defect in rats and compared this to unseeded aorta dECM and surgical mesh. After being implanted for six months, they found a significantly lower number of abdominal adhesions in the animals treated with cell-seeded dECM (the most adhesions were with surgical mesh), and the highest durability and tensile strength was from surgical mesh, but the cell-seeded dECM was not statistically different from the surgical mesh in tensile strength. They also found a significantly higher capillary density and number of fully developed blood vessels, with a reduced amount of inflammation compared to both surgical mesh and unseeded dECM. This suggests that culturing cells on a scaffold prior to implantation might improve the functionality and outcomes of the graft [170].

## 5. Applications of dECM

Given the favourable properties of ECM, the use of dECM in tissue engineering and regenerative medicine has been expanding in many different areas. *In vitro*, dECM is proven to be a useful biomaterial for developing new approaches to tissue engineering for addressing disease as well as the modelling of diseases. *In vivo*, dECM has been shown to be highly biocompatible, be readily utilized in repairing damaged or lost tissues, and promote rapid integration into tissues.

### 5.1. In Vitro Cell Culture with dECM

*In vitro* use of ECM as a scaffold for tissue engineering presents many opportunities. As previously discussed in this paper, dECM as an in vitro substrate has many beneficial effects on cells being cultured, including stabilizing and enhancing the cell phenotype and improving cell viability, proliferation, function, etc. The use of dECM as a material for in vitro study provides the first step towards the potential clinical use of dECM for treating health conditions and for use in pharmaceutical research.

In addition to the study of growing healthy tissues, dECM provides opportunities for studying disease more accurately than in traditional tissue culture approaches, including cancer. In considering the dynamic reciprocity that occurs *in natura* in ECM, two approaches to studying tumour physiology with dECM are to study the effect of the dECM on cells and to study the effect of tumour cells on dECM. Tumour-derived dECM has been shown to influence the behaviour of cells, inducing or enhancing cancer-like behaviours. Helal-Neto E et al. (2016) showed that cell-cultured melanoma-derived dECM has a different composition than melanocyte-derived dECM and that endothelial cells cultured on the melanoma-derived dECM showed increased adhesion, migration, proliferation, and tubulogenesis; and cells were potentially potentiated in a way that would up-regulate angiogenesis [126]. Liu G et al. (2018) examined the effect of human breast cancer–derived dECM on MCF-7 cells (a human breast adenocarcinoma cell line), and showed that the stem cell markers of the cells were upregulated in the tumour-derived dECM over the cells grown as a monolayer, still showing significantly higher stem cell markers than cells in a monolayer following treatment with the cytotoxic chemotherapy agent 5-fluorouracil [127]. Jin Q et al. (2018) compared dECM derived from human breast cancer with healthy breast tissue and found that MCF-7 cells cultured on breast cancer-derived dECM showed an overexpression of epithelial–mesenchymal transition markers and cancer proliferation, whereas dECM from healthy tissue inhibited the epithelial–mesenchymal transition and induced apoptosis that resulted in lower cell growth [129]. In a study by Zhao L et al. (2017), the authors derived dECM from tongues, which they used to co-culture Cal27 tongue squamous cell carcinoma cells and primary cancer-associated fibroblasts, and found the tongue-derived dECM to be favourable to studying the behaviour tongue squamous cell carcinoma [352]. Alabi BR et al. (2019) created an *ex vivo* system to study a human colon cancer cell line in colon dECM from a wild-type and a colon cancer–susceptible mouse colon. They found that the dECM from colon cancer–susceptible mice allowed for increased cell migration, invasion, proliferation, and differentiation than colon dECM from wild-type mice [353]. A study by Mazza G et al. (2019) compared the influence of dECM from healthy and cirrhotic human livers on HepG2 cells (a human liver hepatocellular carcinoma cell line) and found that the dECM from cirrhotic livers increased epithelial–mesenchymal transition and promoted the progression of hepatocellular carcinoma *in vitro* [130]. Kort-Mascort J et al. (2021) used dECM for porcine tongues to create a composite hydrogel with gelatin and alginate, which they 3D-bioprinted to create scaffolds to study chemotherapy treatments on head and neck cancers (as represented by two immortalized human squamous cell carcinoma cell lines). They reported a greatly increased IC50 response to cisplatin and 5-fluorouracil compared to a 2D culture, suggesting that dECM and a 3D culture are important to developing disease models [354]. Though not part of the ECM, nanovesicles collected from the ascites from cancer patients with high-grade epithelial ovarian cancer have shown that the bioactive molecules in nanovesicles have the potential to affect treatment-induced cancer-adaptation processes [355]. This shows the potential for dECM to be used to study cancer in vitro and suggests the possibility of more accurate studies of other diseases and improved disease modelling.

Tissue engineering is being explored as a method of in vitro pharmaceutical research and development, and such research includes the use of dECM. Sasikumar S et al. (2022) used liver dECM to maintain the phenotype of hepatocytes, and a co-culture of hepatocytes with endothelial and stellate cells in liver dECM showed sensitivity to in vitro drug-induced liver injury [106]. Ferreira LP et al. (2021) used breast-derived dECM to develop metastatic breast cancer–fibroblast microtumour models; the microtumour spheroids recapitulated key breast tumour hallmarks, and the cell-laden co-culture and monoculture spheroids could be used for more accurate cytotoxic drug screening assays [356]. Though these are early results, they show the importance of the ECM in developing accurate tissue models for pharmaceutical research and development.

### 5.2. ECM Extract

Using intact or solubilized dECM is not the only way that the ECM is used in tissue engineering research; liquid extracts from dECM are also used to improve cell cultures and promote cell differentiation. The nanovesicles found in the ECM have various cell-signalling molecules (e.g., miRNA molecules, cytokines, etc.), which are usually released during the remodelling or inflammation of tissue. These molecules can trigger immune responses in vitro in macrophages that are analogous to the response seen in the parent ECM as well as promote the faster maturation of neurites than in the parent ECM [23]. The growth media that were used to saturate dECM prior to being used for cell culture have been shown to improve cell viability and metabolism, and enhance differentiation markers [86,91,92,134]. Urea is often used to denature and solubilize proteins when used at high concentration and can be used to prepare an extract of soluble molecules from dECM that can enhance cell proliferation [86].

### 5.3. Clinical Use of dECM

Decellularized ECM has been used to repair or replace tissues in clinical trials. The use of the ECM for medical purposes dates back thousands of years: clear records of catgut and sinew sutures date to the time of the 3rd century CE in the Greek physician Galen’s writings on treating injured gladiators, possibly date even earlier to c. 500 BCE in the writings of the Indian surgeon Susruta [357], possibly date to c.1500 BCE in Ancient Egypt in the Ebers Papyrus (believed to be copied from earlier texts), and likely date much further back in history [53,358]. In much more recent history, decellularized xenogeneic heart valves have been fabricated using dECM, and hollow tissues such as bladders and intestines have been decellularized and used as thin sheets or powder for surgical procedures to promote repair and regenerations in vivo, typically of mucosal tissues, and also more complex tissues such as muscle and fingers [316,359,360,361,362]. These clinical trials using dECM have shown great promise in triggering repair and regeneration of tissues. See Table 5.

Simple sheets of dECM can be manipulated to create custom shapes, allowing the biomaterial to be used in very versatile ways. In multiple clinical trials, these simple sheets have been used to promote the repair and regeneration of tissues. Most clinical research has been done on mucosal tissues, but the sheets have also been used for more complex tissues such as muscle.

In 2011, Badylak et al. published the results of a five-patient trial on treating Barrett’s oesophagus, a type of adenocarcinoma that can develop in the lower oesophagus. The standard treatment has been either endoscopic mucosal resection, phototherapy, or radiofrequency ablation, all of which cause scarring, strictures, or esophageal resection, which have high morbidity and mortality rate. In the trial, the patients underwent endoscopic mucosal resection and then had an dECM scaffold placed at the resection site. Within four months, a nearly complete mature oesophageal squamous epithelium had formed where the scaffold had been placed. Of the five patients in the trial, two patients had a small focal recurrence of Barrett’s oesophagus at the gastroesophageal junction (likely due to the uncorrected hiatal hernias that were the original underlying cause of the dysplasia), but the rest of the reconstituted mucosa remained disease-free. This showed that dECM could be used to treat mucosal injuries [359].

One of the next major successes that came from the Badylak Lab was repairing the volumetric muscle loss in people who had lost large portions of muscle in a trauma. They conducted a trial of 13 patients who had lost an average of 66.2% of tissue in one limb as compared to the contralateral limb. Loss of this much muscle had greatly affected the use of the injured limb and reduced the quality of life for the trial participants. As a part of the trial, the participants undertook 4–16 weeks of physical therapy pre-operatively for them to plateau in performance. The next step was a surgery that involved opening the muscle compartment, debriding the scar tissue at the site of the injury, and performing selective tenolysis where adhesions had formed. Using dECM sheets to construct the scaffold, the sheets were cut to fit the defect, and were placed in contact with the native healthy muscle tissue. Within 24–48 h post-operative, the patients began a 24-week physical therapy regimen to promote remodelling and regrowth. At six months post-operative, on average the participants showed an improvement in strength of 37.3% and in range-of-motion of 27.1%. This trial showed that dECM can be used as a viable treatment for volumetric muscle loss [44,361].

The Badylak Lab is not the only research group working on using dECM to create scaffolds. One of the other major pioneers is the Wake Forest Institute for Regenerative Medicine (WFIRM). WFIRM was involved in a clinical trial using dECM that began in 2005 to engineer vaginas using dECM to treat Mayer-Rokitansky-Küster-Hauser syndrome (MRKHS), a condition where there is congenital vaginal agenesis of the proximal two-thirds of the vagina (there might also be other abnormalities and/or absences of the cervix and uterus, and the condition can affect other systems in the body) [363]. In this pilot study, the researchers constructed vaginas for four teenage female patients with MRKHS using decellularized intestinal submucosa. The procedure involved a full-thickness biopsy of the available vaginal tissue in each patient five to six weeks prior to surgery. The tissue from the biopsy was separated into the muscle and epithelial tissue layers, with each tissue layer being cultured with the respective methods to expand the cell populations. The scaffolds were seeded with cultured cells for seven days, then surgically implanted into the patients (two patients were able to have an anastomosis created with their respective uteri that allowed them to menstruate, though the other two patients had uterine agenesis that prevented this). The engineered vaginas developed into phenotypical and functional vagina-like tissue similar to that of normal vaginal tissue. Assessment using the Female Sexual Function Index questionnaire reported normal values in the six domains of desire, arousal, lubrication, orgasm, satisfaction, and painless intercourse for all four patients. This study demonstrated that it is possible to create functional vaginas using dECM [362].

Other small clinical trials have been conducted, often as n of 1 studies. In one case, dECM sheets were used in conjunction with a stent to repair recurrent anastomotic strictures following a repair of a congenital long-gap oesophageal atresia, and, after two treatments, there was successful growth of new mucosal tissue, with minimal inflammation and no recurrence of the stricture after stent removal [364]. In two separate cases where dECM was used to repair damaged urethras, the grafts appeared to form natural urethral mucosa [365]. Kesavan R et al. (2021) used minimally manipulated dECM from autologous homologous adipose tissue to 3D print a wound patch, and all of the patients in the treatment group showed complete wound closure and epithelization within four weeks, whereas patients in the control group who received the standard wound care showed delayed wound healing (~50% at 12 weeks) [366]. Using dECM from the bladder and small intestine has been shown to work well for treating some mucosal tissues.

Tissue-engineered heart valves would be an important clinical application for dECM. A custom mitral valve was created from dECM to treat a 15-month-old boy who had developed an infectious mass on their native valve. As the child was so young, the surgeons felt that traditional artificial valves were not well-suited, so they removed the infected valve and attached the custom valve graft to the annulus and papillary muscles. Three months after the surgery, the child had no significant mitral stenosis, there was laminar flow through the dECM mitral valve, and there was no significant regurgitation, and the child was at a normal baseline for activity level [367]. See Table 4 for a list of other clinical trials with dECM.

**Table 5 jfb-13-00240-t005:** Clinical trials using dECM that were registered with ClinicalTrials.gov (accessed on 17 October 2022).

Source	Targeted Condition	Phase	Recruitment Status	Outcome	Material	Year Posted	Reference
Porcine small intestine submucosa	Rotator cuff tear	4	Recruiting	N/A	ArthroFLEX ECM scaffold graft	2018	[368]
Fish skin	Chronic wounds	N/A	Completed	Not reported	MariGen Wound Dressing	2011	[369]
Adipose	Obesity	N/A	Completed	Not reported	Adipose allograft extracellular matrix	2016	[370]
N/A	Ischemic cardiomyopathy	1/2	Not yet recruiting	N/A	Wharton’s jelly-derived mesenchymal cells seeded onto an extracellular matrix patch	2019	[371]
Porcine urinary bladder	Neuropathic diabetic foot ulcer	N/A	Completed	Not reported	MatriStem	2016	[372]
Porcine myocardium	Myocardial infarction-induced heart failure	1	Completed	Not reported	VentriGel	2014	[373]
Human dECM	Articular cartilage repair in microfracture surgery	1/2	Recruiting	N/A	HST-003	2021	[374]
Porcine small intestine submucosa	Pericardial reconstruction	N/A	Completed	Elevated pro-inflammatory proteins in blood for all patients (similar to control not treated with CorMatrix) No adverse events for the treatment or control groups	CorMatrix ECM	2014	[375]
Porcine small intestine submucosa	Implantable electronic device placement for cardiovascular diseases	N/A	Completed	Of the 1025 patients in the SECURE trial: 14 had an ECM-related adverse event, possibly related to CanGaroo 2 had an ECM-related adverse event, probably related to CanGaroo 12 had major pocket infections	Cormatrix CanGaroo ECM Envelope	2015	[376]
Porcine small intestine submucosa	Pericardial reconstruction following coronary artery bypass graft surgery	N/A	Terminated	Elevated pro-inflammatory proteins in blood for all patients (similar to control not treated with CorMatrix) No adverse events for the treatment or control groups	CorMatrix ECM	2012	[377]
Porcine small intestine submucosa	Chronic wounds	N/A	Completed	Not reported	Oasis Extracellular Matrix	2018	[378]
Human adipose tissue	Soft Tissue Injuries	1	Completed	Graft demonstrated satisfactory safety results No participants experienced serious nor unanticipated adverse events (all were expected and mild)	Acellular adipose tissue	2016	[379]
Human adipose tissue	Soft Tissue Injuries	2	Active, not recruiting	N/A	Acellular adipose tissue	2018	[380]
Porcine small intestine submucosa	Inguinal hernia repair	4	Completed	Not reported	Surgisis Inguinal Hernia Matrix	2008	[381]
Ovine forestomach	Reconstruction of soft tissues	4	Recruiting	N/A	Myriad Matrix, Myriad Morcells	2022	[382]
Porcine urinary bladder	Pressure ulcer	N/A	Completed	2 of 20 patients had complete wound epithelization at 12 weeks	MicroMatrix ACell Cytal Wound Matrix 2-Layer	2017	[383]
Porcine dECM	Volumetric muscle loss	N/A	Completed	Average improvement of strength: 37.3% Average improvement in range-of-motion tasks: 27.1% No serious adverse events	ACell, Matristem Cook, BioDesign Bard, XenMatrix	2011	[361,384]
Porcine small intestine submucosa Fish skin	Punch biopsy wounds	N/A	Completed	Not reported	Oasis ECM (porcine) MariGen Wound ECM dressing (fish)	2013	[385]

## 6. Future Directions

The goal of tissue engineering is to combine scaffolds, cells, and biologically active molecules to create functional tissues. Currently, there are many different biomaterials, hydrogels, and scaffold designs being researched with the goal of treating and even curing disease through regenerative medicine. Using dECM as a scaffolding material, in combination with cells and biologically active molecules, has shown great potential for clinical use.

There is still much to learn about the ECM, with several potential and important areas of research that future studies should focus on. The ECM is of vital importance to physiology, and the ECM is a highly complex functional and structural biomaterial that constitutes an ideal environment for the reconstruction of healthy, functional tissues. This is because the ECM intrinsically contains many of the biochemical and mechanical cues that regulate cell physiology, including differentiation for specific phenotypes and maintenance of function. The highly conserved nature of the ECM across evolution highlights the importance of the ECM for the physiology and function of cells, tissues, and organs. Moreover, the breadth of research on the use of dECM for tissue engineering with various tissue sources, processing methods, and potential target applications has demonstrated the potential of dECM for disease modelling in vitro and in vivo research and potential use in regenerative medicine for therapies. The insights gleaned from exploring the remaining questions about the ECM will help to lead to new understanding, new techniques, and the achievement of clinically desirable outcomes. Here, we highlight some of the research questions that remain to be answered.

### 6.1. Optimizing Decellularization

Decellularization of tissues can be simple with chemical baths, or more complex with perfusion methods, not to mention other methods of enhancing or directly decellularizing tissues. That detergents such as SDS and Triton X-100 damage the ECM during decellularization highlights the importance of research to find new methods and new chemicals to minimize the exposure to these detergents, if not outright replace them. The diversity of tissue types that could be used for preparing dECM suggests that one strategy will not be amenable to all tissue types. As demonstrated by the diverse methods of decellularization discussed in this review, there are a lot of techniques that could be used or combined with existing methods to speed up the decellularization process, minimize damage to the ECM, ensure appropriate levels of decellularization are achieved, and further improve the biocompatibility of dECM for cell culture and *in vivo* and clinical use.

### 6.2. Stabilizing dECM

A major obstacle to the use of dECM in treating pathologies or culturing cells is that dECM is not a durable material, with dECM hydrogels having even less durability than intact dECM. Before dECM can find widespread use, the durability will need to be improved. Tailoring the degradation rate of dECM to match the rate at which tissue is growing is important for optimal outcomes. Degrading too slowly can lead to fibrosis, but degrading too rapidly means the tissue cannot grow properly. Although dECM sheets have been used clinically with good success in certain applications, finding ways to evenly and accurately enhance the durability of intact dECM, especially in dECM derived from solid 3D organs, in a manner that allows for tuning the degradation rate will provide new opportunities for reseeding decellularized organs. Combing solubilized dECM with other polymers to form a hybrid hydrogel with tunable degradation rates is an important area of research for the future of dECM use. Optimizing the pre-cellularization of dECM prior to use *in vivo* has the potential to better stabilize dECM, while also promoting the rapid repair and regeneration of tissues. Establishing the proper degradation rate of dECM for specific tissues and then developing methods for stabilizing dECM for *in vivo* use will be an important step to expanding research and clinical use.

### 6.3. Fabricating Custom Scaffolds

The 3D *in natura* architecture of tissues is incredibly detailed and very precise, to ensure that the tissues are functioning optimally. While decellularizing solid organs allows for this architecture to be preserved in the ECM that remains, this does not permit the customization of a scaffold for specific needs and for potential future clinical use as grafts and transplants. One potential way to a create custom scaffold from dECM is to use additive manufacturing techniques such 3D bioprinting and bioplotting with fused deposition modelling. By combining different tissue-specific dECM-based inks, a specific anatomical structure can be replicated. This will require developing bioinks from different tissue sources to allow for the specific placement of target cell types for a tissue as well as bioreactors for culturing the cells in the scaffold. Advanced methods, such as electrospinning and coaxial printing, can create scaffolds with engineered properties and features that can be used for creating placement of specific anatomical features within a scaffold. Methods of sterilization of dECM and especially scaffolds made from solubilized dECM hydrogels, are necessary to move this research forward. Ultimately, creating a scaffold that is capable of supporting the growth of tissues over the long term, which can be implanted and provided with a blood supply, whether through some type of anastomosis with native vasculature or angiogenesis from neighbouring vasculature, is still a largely unmet goal in tissue engineering.

### 6.4. Recellularization

A key challenge of tissue engineering is the cellularization of scaffolds, including the recellularization of dECM scaffolds. The selection of the appropriate cell type and seeding method will depend on the tissue type and scaffold design being studied. For simple dECM scaffolds, a simple incubator is sufficient, but more complex scaffolds, especially decellularized organs with an intact vascular tree, require more complex bioreactor systems. These bioreactor systems are vital for the production of functional tissues and organs as they provide the dynamic culture conditions that mimic *in natura* conditions, which help to stimulate and maintain the specific physiological conditions for cells to function for the given tissue. Each organ and tissue type will have specific considerations for recellularization, especially in organs with a complex arrangement of cells of different types, which include the different routes to seed cells and ensure proximity to the nutrient supply and effective elimination of metabolic wastes. Developing bioreactor systems specifically for recellularizing complex organs and scaffolds fabricated with solubilized dECM hydrogels is important for the production of functional organs.

### 6.5. Mechanism of Action of ECM

ECM-based biomaterials are able to elicit a wide variety of favourable responses, both *in vitro* and *in vivo*, that promote healthy, functional tissue reconstruction; however, the mechanism by which the responses are achieved, including stem cell recruitment and angiogenesis, innate antimicrobial activity, and modulation of the immune response, are still generally unknown. One or more mechanisms might play a role, and these mechanisms are believed to include biomechanical and biochemical signals, the cell interaction between the integrins and the ligands in the ECM, the release of various biological signalling molecules embedded in the ECM, the creation of crypteins and exposure of cryptic epitopes, and the variation between tissue types of proteins and other structural biomolecules. Developing a better understanding of the biology and regulating effects of the niche microenvironment of the ECM in the remodelling and repair response will help to advance research in the fields of tissue engineering and regenerative medicine.

### 6.6. Clinical Use of dECM for Regenerative Medicine

As research expands the potential applications and techniques for using dECM to improve methods of tissue engineering, it will also expand to increase the related clinical translation. The early stages are already happening, with the use of simple dECM materials (i.e., powders and sheets) being used in clinical trials and being used as part of available treatment modalities. More complex tissue-engineered scaffolds are being developed and reaching the bedside in preliminary clinical trials, but this is still in the earliest stages of translational research. As research in tissue engineering develops better methods of growing constructed tissues into complex organs, there will be opportunities to use these organoids for pharmaceutical research, both in modelling diseases for the research and development of new pharmaceutical therapies and as a step in the processes of regulatory approval. More complex organoids have the potential to be used to augment failing organs as a clinical treatment option for patients. An ultimate goal is the ability to construct an organ for transplant into a patient that will revolutionize the treatment of a chronic disease and free people from a long wait on the transplant waitlist. These engineered tissues will require the ECM, whether the scaffold is made from dECM or if the cells replace the scaffold with ECM, and additive manufacturing techniques and the recellularization of dECM are two very promising methods for this.

## 7. Conclusions

An important consideration at the start of research in tissue engineering and regenerative medicine is how the research will have direct or indirect clinical relevance, so the hypothesis being evaluated needs to take into account the conditions present in physiological environment. Tissue engineering applies the triad of base sciences—biology, chemistry, and physics—to engineer solutions to reconstruct the human body. The scaffolds used in tissue engineering seek to provide an ECM substitute to support cells in culture, but over millions of years of evolution, nature has produced a very complex material with intricate functions that is hard to replicate. Our understanding of the ECM is evolving, with new discoveries being made every day, which moves the technology forward, improving the lives of patients being treated with these strategies. Using dECM as the scaffold material has benefits over other natural and synthetic materials because it already possesses much of the bioactivity and properties of native tissue. However, there are still many unknowns, and a lot left to understand about the biochemical and structural nature of the ECM, so a multidisciplinary approach is required to sleuth out the answers to these challenges. Overall, great progress has been made in using the ECM for applications in tissue engineering and regenerative medicine, including successful clinical translation for some applications, demonstrating the tremendous clinical potential and feasibility of a wider use for dECM, as the research translates from the bench to the bedside. The role of the ECM in the exponential technology revolution happening in tissue engineering and regenerative medicine is vital to success. Knowledge about ECM has come a long way from the ancient developments in using this material for making tools and clothing, to the early medical uses as sutures, to the modern research on the use of the ECM for tissue engineering, and the future of the ECM for tissue engineering and regenerative medicine looks bright.

## Figures and Tables

**Figure 1 jfb-13-00240-f001:**
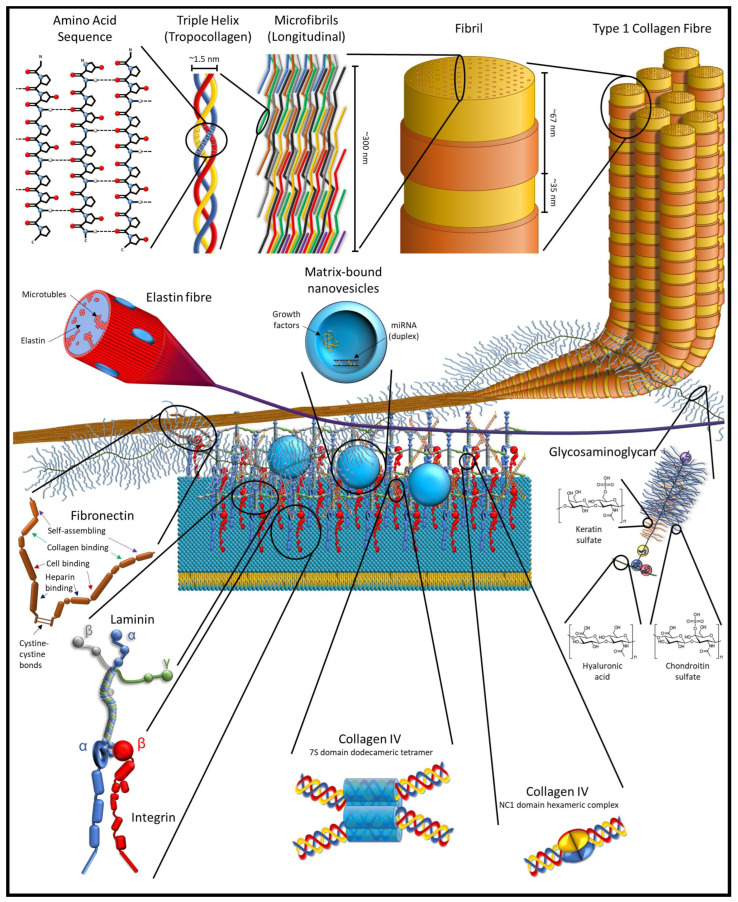
Artistic representation of the structural arrangements of major proteins in the ECM. Structures are not drawn to scale.

**Figure 2 jfb-13-00240-f002:**
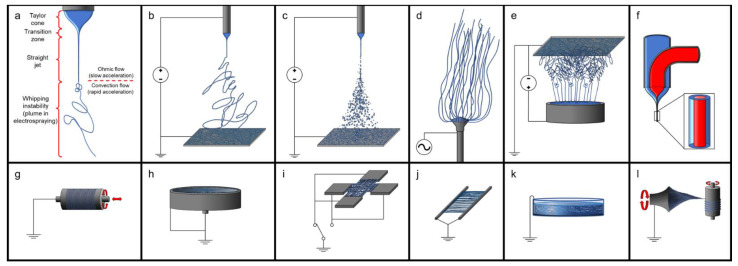
Electrospinning and electrospraying. (**a**) Diagram of the section of electrospinning process; (**b**) typical electrospinning setup depicted with an electrified spinneret, a grounded plate collector, and a high-voltage DC power source; (**c**) typical electrospraying setup depicted with an electrified sprayer, a grounded plate collector, and a high-voltage DC power source; (**d**) typical electrospinning set-up depicted with an electrified spinneret, a grounded plate collector, and a high-voltage AC power source; (**e**) a form of needleless electrospinning set-up depicted with an electrified sprayer, a grounded plate collector, and a high-voltage DC power source; (**f**) coaxial electrospinning/electrospraying; (**g**–**l**) other options for electrospinning collectors include (**g**) rotating and rastering mandril collector (rastering is optional), (**h**) cup-and-pin collector, (**i**) perpendicular collector, (**j**) parallel collector, (**k**) wet electrospinning, and (**l**) yarn electrospinning.

**Table 2 jfb-13-00240-t002:** Methods of decellularization.

Method	Types	Mechanism	References
Chemical			
Detergents			
	Ionic	Solubilize cell and nucleic membranes, denature proteins, and remove cell debris; can disrupt ECM structure and remove desirable biological molecules (e.g., GAG, growth factors, etc.)	
	Sodium dodecyl sulfate		[82,87,88,90,93,109,110,111,112,114,115,116,117,118,128,134,135,136,137,143,144,146,148,151,152,153,154,155,156,157,158,159,160,161,162,163,164,165,166,167,168,169,170,171,172,173,174,175,176,177,178]
	Sodium deoxycholate		[56,113,123,138,151,179,180,181,182,183,184,185,186,187]
	Sodium lauryl ether sulfate		[127,149,188,189]
	Sodium lauroyl sarcosinate		[125]
	Potassium laurate		[190]
	Triton X-200		[172,191,192]
	Nonionic	Disrupt DNA, lipid, and protein interactions; renatures proteins; less effective than SDS	
	Triton X-100		[14,23,82,83,86,88,90,96,97,107,109,110,113,114,115,117,123,133,134,135,136,137,138,139,144,151,152,153,154,155,156,157,158,159,160,161,162,173,174,177,178,179,180,181,182,183,186,187,193,194,195,196,197,198,199,200,201,202,203,204,205]
	Zwitterionic	Exhibit properties of non-ionic and ionic detergents; minimal disruption of ECM structure	
	CHAPS		[198]
	Sulfobetaine-10 and -16		[191,192]
	Amidosulfobetaine-14		[167,206]
Acid/Base			
	Acids	Denature proteins, solubilize the cytoplastic contents of the cell, and degrade genetic material	
	Peracetic Acid		[131,173,207]
	Base		
	Sodium hydroxide		[208]
	Ammonium hydroxide		[14,90,139,155,172,176,204,205,209]
	Buffer		
	Tris	Extract soluble cytoplasmic proteins, increase permeability of cell membranes, and degrade genetic material	[20,23,83,97,116,128,151,158,166,167,194,197,199,200,202,210]
Other			
	Hypotonic/hypertonic		
	Sodium chloride	Osmotic shock that can cause lysis or shrinking of cells	[143,172,180,185,199]
	Potassium iodide		[134,206]
	Potassium chloride		[134,206]
	Chelating		
	EDTA, EGTA	Bind to metal ions that have the potential to interfere with the activity of enzymes used in decellularization, disrupt cell adhesion to the ECM, and inhibit metalloproteases	[14,20,23,83,84,86,91,107,111,112,116,128,131,133,134,135,137,138,143,154,155,166,172,186,187,194,195,196,197,198,199,200,202,211]
	Penetration enhancement		
	DMSO	Protect ECM structure, increase the penetration of detergents, and shorten decellularization time	[151,166]
Biological			
	Enzymes		
	DNase	Degrade genetic material through hydrolysis, leading to a reduction in the fragment size	[56,82,83,86,88,93,96,112,113,115,116,128,132,134,139,154,156,158,166,170,172,175,179,180,181,183,184,194,198,201,204,205,210,212]
	RNase		[56,82,112,132,156,158,179,180,181,183,194,204,205,210,212]
	Benzoase		[123]
	Trypsin	Digest proteins and help to disrupt cell attachment to the ECM	[23,107,112,115,135,138,155,186,187,196,197,202,210,212,213]
	Non-enzymatic agents		
	PMSF	Inhibit proteases that are released during cell lysis that have the potential to damage the ECM	[20,86,158,199]
Physical			
Freezing		Disrupt cell membranes due to ice crystal formation and expansion	[20,56,82,84,86,87,88,93,109,111,112,113,116,125,132,133,134,135,137,153,154,155,163,169,179,180,182,183,197,199,200,210,211,214,215]
Heating		Denature cell proteins, inactivate enzymes, disrupt cell membranes	[103]
Sonication		Aid in cell lysis and removal of cell debris	[216,217]
Electroporation		Disrupt cell membranes	[218,219,220,221,222]
Vacuum		Cause cell lysis, improves penetration of decellularization solutions	[118,146,180,183]
Mechanical		Physically remove unwanted tissue layers to allow more effective decellularization	[23,92,158,195,200,223,224,225,226,227]
Techniques of applying agents			
Perfusion		Thoroughly deliver decellularization fluid to all parts of a tissue	[90,117,123,135,137,138,139,144,148,152,153,157,164,165,169,193,214,228,229]
Agitation		Ensure adequate mixing of solutions used in decellularization and aid in cell lysis and removal of cell debris	[20,23,86,88,116,125,127,128,130,133,134,146,153,158,167,180,181,182,194,196,200,210,223,227,230]
Supercritical fluid		Disrupt the cell membrane and cause cell lysis with a minimal effect on the ECM and ECM components	[201,231,232]

**Table 3 jfb-13-00240-t003:** Summary of advanced decellularization methods (results are compared to standard decellularization protocols).

Method	Purpose	Results	References
Vacuum	Enhanced penetration of decellularizing solution and clearance to cell debris	Effective at removing DNA, MHC-1, and other cellular content Faster and more efficient decellularization Minimal effect on collagen, GAG content, and biomechanical properties Potential to damage ECM microstructures in weaker tissues	[118,146,180,183,263]
Hydrostatic washing	Enhanced penetration of dense, fibrous tissues	Faster and more efficient decellularization	[264]
Pulsatile perfusion	More closely mimics *in natura* state of tissue perfusion	More profound decellularization More homogeneous decellularization Lower residual DNA content Little to no difference in collagen and GAG content	[165,245]
Chemical penetration enhancement	Improve solubility and penetration of detergents and disrupt cell membranes	Faster and more efficient decellularization Reduced DNA content Better preserved and protected GAG, elastin, and collagen	[166]
Sonication	Disrupt cell membranes due to cavitation	Minimal effect on the fibrous structure of ECM More effective at removing cells Greatly reduced the decellularization time	[216,217]
Nonthermal irreversible electroporation (NTIRE)	Cause irreversible damage to the cell membrane	No effect on the ECM structure Causes tissue disruption, cell delamination, and cell death	[165]
Decellularizing agents	Different levels of reactivity with biological molecules in addition to disruption of cell membranes and clearance of cellular debris	Potassium laurate Better retention of ECM compounds Better-preserved architecture Increased cell viability and proliferation in vitro Lower inflammatory response and better cell distribution *in vivo*	[127,149,188,190,208]
		Sodium lauryl ether sulfate Better GAG and collagen retention Better preserved microarchitecture Lower inflammatory response and platelet adhesion *in vivo* Better host cell migration into scaffold Slower at decellularizing	
		NaOH As effective as detergents Similar collagen, GAG, and adhesion protein retention Better DNA clearance Equal to detergents for *in vivo* cell migration	
Supercritical fluids	Enhance penetration of cell membrane and clearance of cell debris	Comparable to detergents in cell and DNA removal Superior retention of GAG, soluble collagen, adhesion proteins, and angiogenic factors Need to add collagen to gel for gelation to occur Superior neovascularization No difference in immune response	[201,232]
Alternating decellularizing solutions	Minimize exposure time to detergents in combination with hypertonic and hypotonic shock to cells, clearance of detergents, and clearance of cell debris	Slower decellularization Equal DNA removal Superior retention of GAG and soluble collagen Similar retention of growth factors Higher cell viability and gene expression profiles *in vitro*	[228]

**Table 4 jfb-13-00240-t004:** Sterilization techniques for dECM.

Method	Pros	Cons	References
Peracetic acid and ethanol (together or separately)	Minimal effect on structural and biological properties of ECM Peracetic acid is a strong oxidizer with effective bactericidal, viricidal, fungicidal, and sporicidal properties Can be used to decellularize tissues in combination	Can be difficult to remove Residual chemicals can negatively affect cell viability	[23,83,92,93,109,114,135,136,138,144,155,158,159,161,162,169,179,182,187,189,200,211,223,225,226,227,230]
Ethylene oxide	Effective at sterilizing biomaterials	Residual gas can negatively affect cell viability Minimally effective in hydrogels Can affect the mechanical properties of the final hydrogel	[195,224,236,347]
UV	Can improve mechanical properties due to cross-linking	Harmful to cells Can cause premature gelation of dECM hydrogels	[118,133,153,169,211]
Antimicrobials	Can be added to culture media Can be used during decellularization to limit contamination during processing	Can be difficult to remove Can affect cell behaviour	[112,113,116,117,127,133,135,137,144,148,152,154,156,164,166,167,168,179,184,193,214]
γ radiation	Highly effective in reducing bioburden No effect on cell response	High levels of exposure can prevent gelation of dECM hydrogels	[158,194,195,347]
Electron beam (β radiation)	Highly effective in reducing bioburden	High levels of exposure can prevent gelation of dECM hydrogels	[195,348]
Sterile filtration	Can be used to filter ECM extracts	Loss of proteins in high concentration dECM solutions and colloidal dispersions	[86,91]
Supercritical CO_2_	Highly effective at reducing bioburden Can be combined with other sterilizing agents (e.g., ethanol, peracetic acid)	Can affect the mechanical properties of the final hydrogel	[195,201,231,349,350]

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
