# Peer review of "Preparation and Use of Decellularized Extracellular Matrix for Tissue Engineering"

_jfb, 2022, doi:10.3390/jfb13040240_

Round 1

Reviewer 1 Report

In this article, the author discusses the preparation method of ECM-based biomaterials and its applicable areas as promising biomedical building blocks. The article covers a wide range of topics which are crucial concerns of scientists in applying ECM-based biomaterials in the field of tissue engineering and regenerative medicine.

Several specific issues which need to be addressed according to the manuscript are listed as follows.

Major concerns

1.     This title of this review article is rather broad and would be better suited for a book chapter on extracellular matrix and tissue engineering. It is advised to focus the title on the central topics that the article tends to develop.

2.     Given the abundant availability of review articles on extracellular matrix-related subjects, this topic of this review is not particularly original. The abstract and introduction sections cannot adequately convey the novelty of the entry point.

3.     The hierarchical assembly of collagen I fibrils in the ECM, which is depicted in the article along with its structure and properties, is illustrated in figure 1. Figure 1 needs to be updated to reflect how the entire ECM is organized in this circumstance. Furthermore, in all other reviews, the collagen I hierarchical structure is well-known.

4.     To prompt reflections on how to select the best strategies for various circumstances, the columns in Table 1 should sum up the advantages and disadvantages of employing the techniques indicated there. In addition, different tissues and organs require distinct decellularizing techniques, which should be noted as well.

5.     The “3. Methods of preparing ECM” section is divided into four sections, but I don't perceive any logical coherence between any of them, particularly sections 3.2 and 3.3.

6.     The fourth section, which covers ECM modification and methods, includes strategies for enhancing ECM functional capacity. However, 4.3 summarizes how to solubilize ECM for bioink, which does not follow the main theme logically.

7.     In table 3, one of the pros listed as “one of the most common methods used for sterilizing ECM” might not be the most accurate benefit when expressing how effective this method is. The pro should be precise, in other words.

8.     In section 4.5, the author describes the seeding of cells into the extracellular matrix (ECM), although this section is largely devoted to classifying stem cells, even though the cells that are seeded may not be stem cells. The author then addresses decellularization and recellularization of organs using Ott Lab as an example, however it seems more like a progress recap without any methodological evaluation.

9.     The 4.6 section's main premise is hazy; it reads more like a separate paragraph.

10.  More research should be added in the section of 4.6 to support the central idea.

11.  The title of the fifth section is too broad, which is recommended to be narrowed down to a precise field in the application of ECM in tissue engineering.

12.  As the holy grail in tissue engineering and regenerative, clinical use of ECM is crucial. Hence, it’s recommended to add a table summarizing clinical trials, which contains source, targeted condition, recruitment status, outcome, institute/company, and link/reference.

13.  Figure 2 does not support the primary argument made in the entire essay. To demonstrate why the authors' list of techniques is essential, more information is required.

14.  Future directions of ECM application and clinical translations are not completely defined. It is advised to provide a potential landscape of ECM-based tissue engineering and regenerative medicine.

15.  Although the article did not go into detail, decellularized ECM-based bioink used in 3D bioprinting is a prevalently relevant field in regenerative medicine.

16.  Since there are already commercialized dECM-based products, it is necessary to compare emerging trends with existing ones to perform further translations.

Minor concerns

1.      In line 795, an “a” needs to be deleted.

2.      In line 924, “a” should be altered to “are”.

3.      There is a typo in line 1082, “if” should be replaced to “is”.

4.      In line 1083, “proving” should be replaced to “proven”.

5.      In line 1194, a word is missing after “the”.

Author Response

Thank you for taking the time to thoroughly read and review our manuscript and for providing constructive criticism on how we can improve our paper. We have provided a reply to your comments below and attached an updated copy of the paper.

  1. This title of this review article is rather broad and would be better suited for a book chapter on extracellular matrix and tissue engineering. It is advised to focus the title on the central topics that the article tends to develop.

As this review covers such a wide range central topics throughout, developing a comprehensive title that covers all of this would be very challenging. As this this review covers broad areas of decellularization of ECM and how the decellularized ECM is used, we propose a new title as "The Preparation and Use of Decellularized Extracellular Matrix for Tissue Engineering".

  1. Given the abundant availability of review articles on extracellular matrix-related subjects, this topic of this review is not particularly original. The abstract and introduction sections cannot adequately convey the novelty of the entry point.

As indicated, we have significantly revised the abstract and introduction sections to convey or highlight the novelty of this review. We would see the contribution or novelty of this review mainly rests on (1) a comprehensive, yet critical, review of ECM, from its preparation and modification to its in vitro, in vivo, and clinical applications, along with a number of emerging techniques and trends including 3D printing and (2) the identifications of challenges and issues involved in these fields, along with the discussion of our recommendations for future research.

  1. The hierarchical assembly of collagen I fibrils in the ECM, which is depicted in the article along with its structure and properties, is illustrated in figure 1. Figure 1 needs to be updated to reflect how the entire ECM is organized in this circumstance. Furthermore, in all other reviews, the collagen I hierarchical structure is well-known.

The hierarchical structure of collagen I is discussed in some literature reviews of ECM, though it is more common in reviews specific to collagen. However, in many of these reviews, the hierarchical structure of collagen I is discussed but not often shown graphically, let alone with the how the microfibrils are woven together in fibrils. The figure has been updated to include more of the ECM components, but as there are over 300 types of proteins in the ECM, we selected the more prominent components and structures of the ECM.

  1. To prompt reflections on how to select the best strategies for various circumstances, the columns in Table 1 should sum up the advantages and disadvantages of employing the techniques indicated there. In addition, different tissues and organs require distinct decellularizing techniques, which should be noted as well.

Table 1 was getting too large to fit on a page for publication, which is why this layout was chosen. Within the mechanism column, the advantages and disadvantages are included for some of the techniques, but not all methods have distinct advantages and disadvantages to list based on the limited research available in some topics. Adding an additional two columns to discuss advantages and disadvantages decreases the readability of the tables and creates unused space.

  1. The “3. Methods of preparing ECM” section is divided into four sections, but I don't perceive any logical coherence between any of them, particularly sections 3.2 and 3.3.

Each subsection of section 3 highlights different methods of preparing ECM. Each section is distinct from the other section, which is why they are standalone sections. Within section 3.2, the history of the development of perfusion decellularization techniques is told using narrative approaches before expanding into a broader discussion of the how other researchers have prepared ECM from organs through decellularization. For 3.3, we found limited information about using cell culture to prepare ECM, and what we did find was provided. Because each approach to sourcing and decellularization of ECM presents unique challenges that require different techniques, and the objectives of the resulting ECM are different, they were combined in one section and divided according to the broad categories to highlight these differences.

  1. The fourth section, which covers ECM modification and methods, includes strategies for enhancing ECM functional capacity. However, 4.3 summarizes how to solubilize ECM for bioink, which does not follow the main theme logically.

The intention of include 4.3 within modification and methods is because some researchers modify the ECM into a bioink for use in 3D printing/plotting. This enhances the functional capacity of ECM in the sense that it expands the ways that ECM can be used in tissue engineering to fabricate scaffolds, which is the same theme as the other subsections in section 4.

  1. In table 3, one of the pros listed as “one of the most common methods used for sterilizing ECM” might not be the most accurate benefit when expressing how effective this method is. The pro should be precise, in other words.

We agree, and we have changed this and other points in the table.

  1. In section 4.5, the author describes the seeding of cells into the extracellular matrix (ECM), although this section is largely devoted to classifying stem cells, even though the cells that are seeded may not be stem cells. The author then addresses decellularization and recellularization of organs using Ott Lab as an example, however it seems more like a progress recap without any methodological evaluation.

The second paragraph of section 4.5 discusses the classification of stem cells because the first paragraph discusses the use of stem cells and immortalized cells lines, and different types of cells are mentioned throughout the section. In the examples from Ott Lab are used to provide context to the later discussion in the section, as well as some methodological evaluation of the techniques that were used.

  1. The 4.6 section's main premise is hazy; it reads more like a separate paragraph.

Because 4.6 relates closely to 4.3, we have moved 4.6 to be 4.3.1., and reworked some of the paragraphs, which we hope also helps to address your concerns in comment 6.

  1. More research should be added in the section of 4.6 to support the central idea.

This subsection has been expanded to include more research, which we hope also helps to address your concerns in comment 9.

  1. The title of the fifth section is too broad, which is recommended to be narrowed down to a precise field in the application of ECM in tissue engineering.

We have changed the title of this section to "Applications of dECM"

  1. As the holy grail in tissue engineering and regenerative, clinical use of ECM is crucial. Hence, it’s recommended to add a table summarizing clinical trials, which contains source, targeted condition, recruitment status, outcome, institute/company, and link/reference.

We had discussed the results of some clinical trials that used ECM, but we agree that adding more details about clinical trials would be valuable. A table as you suggested has been added to subsection 5.4 to discuss the trials listed on ClinicalTrials.gov that have used ECM.

  1. Figure 2 does not support the primary argument made in the entire essay. To demonstrate why the authors' list of techniques is essential, more information is required.

Figure 2 was added at the request of other readers because they were not familiar with electrospinning and electrospraying. The figure was created to visually represent some of the techniques discussed in subsection 4.2.1. Subsection 4.2.1 has been updated to include more research. The caption for Figure 2 has also be corrected.

  1. Future directions of ECM application and clinical translations are not completely defined. It is advised to provide a potential landscape of ECM-based tissue engineering and regenerative medicine.

We have added a section (6.6) on the future of clinical use of ECM for regenerative medicine.

  1. Although the article did not go into detail, decellularized ECM-based bioink used in 3D bioprinting is a prevalently relevant field in regenerative medicine.

To address this, we have expanded subsection 4.3.1 (formerly subsection 4.6)

  1. Since there are already commercialized dECM-based products, it is necessary to compare emerging trends with existing ones to perform further translations.

We agree. Unfortunately, we were able to find only a very limited number of published research articles on using commercialized dECM-based products in a clinical setting, making a full analysis difficult. We discussed how the details of those clinical trials in subsection 5.4, as well as highlighting the future directions section on how to improve ECM for clinical use in section 6.

Reviewer 2 Report

In this review, the application of decellularized tissue for tissue engineering is reviewed in detail. Decellularized method is important and summarized in this review. Many papers of decellularized method by high hydrostatic pressure have reported, so, it should be included. In this review, does ECM mean decellularized ECM? If so, I recommend including decellularized extracellular matrix in title and abstract because reader maybe confuse natural ECM and decellularized ECM.

Author Response

Thank you for taking the time to read and review our manuscript and for providing constructive criticism. We have attached a revised version of our manuscript.

Many papers of decellularized method by high hydrostatic pressure have reported, so, it should be included.

We had included a mention of hydrostatic washing in subsection 3.4.2, but it was minimal. We have expanded this to include more discussion of research conducted using this method.

In this review, does ECM mean decellularized ECM? If so, I recommend including decellularized extracellular matrix in title and abstract because reader maybe confuse natural ECM and decellularized ECM.

This paper is about the use of decellularized ECM, and it could well be confusing for readers to refer to it as ECM. We have changed the title to "The Preparation and Use of Decellularized Extracellular Matrix for Tissue Engineering" and switched ECM to dECM (decellularized ECM) where appropriate in the manuscript.

Reviewer 3 Report

This is a great reveiw of ECM and tissue engineering and can be published after careful rereading regarding grammar and spelling (only few).

Author Response

Thank you for taking the time to read and review our manuscript and for you kind words on our work. We have reviewed the manuscript for grammar and spelling, and have made changes based on your comments and the comments of other reviewers. We are attaching a copy of the revised manuscript.

Round 2

Reviewer 1 Report

The coverletter file is not the right one and I can not see the tracking changes.